# Psychological stress impairs IL22-driven protective gut mucosal immunity against colonising pathobionts

Christopher R. Shaler[1,2,4], Alexandra A. Parco[1,2,4], Wael Elhenawy [1,2], Jasmeen Dourka[1,2], Jennifer Jury[3], Elena F. Verdu [3] & Brian K. Coombes [1,2,3 ✉]

Crohn's disease is an inflammatory disease of the gastrointestinal tract characterized by an aberrant response to microbial and environmental triggers. This includes an altered microbiome dominated by Enterobacteriaceae and in particular adherent-invasive *E. coli* (AIEC). Clinical evidence implicates periods of psychological stress in Crohn's disease exacerbation, and disturbances in the gut microbiome might contribute to the pathogenic mechanism. Here we show that stress-exposed mice develop ileal dysbiosis, dominated by the expansion of Enterobacteriaceae. In an AIEC colonisation model, stress-induced glucocorticoids promote apoptosis of CD45+CD90+ cells that normally produce IL-22, a cytokine that is essential for the maintenance of ileal mucosal barrier integrity. Blockade of glucocorticoid signaling or administration of recombinant IL-22 restores mucosal immunity, prevents ileal dysbiosis, and blocks AIEC expansion. We conclude that psychological stress impairs IL-22-driven protective immunity in the gut, which creates a favorable niche for the expansion of pathobionts that have been implicated in Crohn's disease. Importantly, this work also shows that immunomodulation can counteract the negative effects of psychological stress on gut immunity and hence disease-associated dysbiosis.

[1] Department of Biochemistry and Biomedical Sciences, McMaster University, Hamilton, ON, Canada. [2] Michael G. DeGroote Institute for Infectious Disease Research, Hamilton, ON, Canada. [3] Farncombe Family Digestive Health Research Institute, Hamilton, ON, Canada. [4] These authors contributed equally: Christopher R. Shaler, Alexandra A. Parco. ✉email: coombes@mcmaster.ca

Crohn's disease (CD) is an inflammatory disorder of the gastrointestinal tract triggered by microbial and environmental insults[1–3]. The global burden of disease associated with CD is rising, particularly in developed countries where an upward trend in incident cases has occurred for decades. Emergent disease is now appearing in Asia, Africa, and South America[4]. Current standards of care, including immunomodulatory biologics, are expensive and have high rates of primary and secondary non-responsiveness. Thus, there is an urgent unmet clinical need to better understand the microbial and environmental triggers of CD that will underpin new preventions and therapies.

The CD-associated microbiome has been intensely scrutinized as a source of inflammation in the gut[5–7]. General microbial features in CD include decreased community diversity, reduced levels of Clostridiales, and increased abundance of Proteobacteria[7–9]. Clinical observations consistently show bacteria in close association with the mucosal epithelium in Crohn's patients, particularly members of the Enterobacteriaceae that are enriched in virulence and secretion pathways[10–12]. Adherent-invasive Escherichia coli (AIEC) is an abundant pathobiont at inflamed sites in the gut[13,14]. Numerous studies have confirmed that AIEC are enriched in humans with CD compared to healthy subjects and are often the dominant bacterial species present[15,16]. AIEC have a multiphasic lifestyle and can grow as extracellular planktonic cells, in biofilms, and intracellularly in epithelial cells and macrophages, where they induce host inflammatory pathways[17–20]. Thus, understanding the colonisation dynamics of AIEC is expected to yield insights into the progression of CD and may inform therapeutic intervention.

Several components of mucosal immunity are disrupted in CD. These include epithelial barrier integrity, mucous production and cell turnover, the production of antimicrobial peptides and proteins (AMPs), and the release of metal ion scavengers that contribute to nutritional immunity[21–23]. Nutritional immunity functions to limit bacterial growth by sequestering essential nutrients and metals required for bacterial replication[22]. Collectively, these elements of mucosal immunity work to spatially restrict bacteria largely to the gut lumen and to reduce the overgrowth of pathogens[24]. Interleukin (IL)-22 is a cytokine that regulates several aspects of mucosal immunity in the gut, including the production of AMPs and the activation of nutritional immunity[25,26] Some enteric pathogens including Salmonella and Citrobacter can overcome IL-22-dependent host defenses through AMP resistance and metal acquisition systems that bypass the host defense proteins lipocalin (Lcn2) and calprotectin[27,28]. Similarly, IL-22 has been implicated in the induction of other iron scavengers, including hemopexin and haptoglobin, that can control the enteric pathogen Citrobacter rodentium[29]. The role of IL-22 in host control of CD-associated pathobionts has not been explored.

Disease expression in CD patients follows a relapsing and remitting course, where relatively asymptomatic periods are followed by heightened inflammation and disease activity[30,31]. A considerable body of clinical literature establishes psychological stress as a disease modifier in CD[32–36]. Episodes of acute psychological stress are associated with flares, relapse, and increased inflammatory markers in both the serum and mucosa of patients with CD and ulcerative colitis[33,34,37]. Rodent models of stress have extended these findings, demonstrating increased inflammation[38], barrier disruption[38–40], reactivation of disease induced by chemical exposure[38,41], and increased susceptibility to enteric infection[42]. A previous study demonstrated that chronic stress coupled with DSS-induced colitis resulted in a modest alteration in inflammation and microbiome composition in the cecum and colon[43]. However, the mechanisms driving the mucosal and microbiological dysfunction associated with psychological stress remain unknown, particularly in the ileum.

Here we show in a pre-clinical model, designed to investigate the comorbid effects of psychological stress on microbial composition and immune pathways in the gut, that stress induces a multidimensional loss of host protection, provoking a dysbiotic shift in the ileal microbiome dominated by AIEC and other Enterobacteriaceae, with attendant immunological and barrier defects. These mucosal deficits are traced back to an apoptotic depletion of CD45[+]CD90[+] cells resulting in the loss of protective IL-22 signaling that blunts AMP responses and barrier repair. The combined effects of nutritional immunity and stress-induced impairment of protective IL-22 responses create a favorable niche for AIEC expansion. These deficiencies are correctable using either exogenous IL-22, or by blocking stress-induced glucocorticoid signaling, which restores mucosal immunity and averts ileal dysbiosis including the expansion of AIEC. These data indicate that immunomodulation can normalize the dysbiotic shifts associated with CD in response to psychological stress, providing avenues for therapeutic interventions.

## Results

**Psychological stress promotes the expansion of ileal Enterobacteriaceae.** Psychological stress has a negative effect on disease activity in CD patients[32,33,35], who commonly have decreased bacterial diversity and increased abundance of mucosal-associated E. coli in the gut[7,11,15]. To study the effects of stress on the microbiome, we developed a pre-clinical model of acute psychological stress and implemented this model in conventional SPF mice. Groups of mice were either uncontrived, or deprived of food and water overnight, or exposed to overnight restraint stress[44,45]. Following release from stress, we profiled the microbiome in the ileum, cecum, and colon using 16S rRNA sequencing and compared it to mice deprived only of food and water, which served as the relevant control for restraint stress (Fig. 1a). Following stress exposure, Enterobacteriaceae became a dominant member of the microbiota in all gut regions, whereas it was present at <1% abundance in all unstressed control mice (Fig. 1b–d). Enterobacteriaceae expansion was most profound in the ileum where it accounted for over 80% of the 16S sequences. At the genus level, Escherichia-Shigella had the greatest proportional change in relative abundance following stress, regardless of the intestinal compartment evaluated (Fig. 1e–g, Supplementary Fig. 1). In addition to Escherichia, enrichment of Enterococcus, Proteus, and Mucispirillum were also seen in the psychological stress groups, but not in controls (Fig. 1e–g). This microbial profile is consistent with that of CD patients and mice with spontaneous ileitis[7–9,46]. Bacterial genera with typically lower abundance in CD or in murine models of colitis include Lachnospiraceae, Bifidobacterium, Faecalibaculum, and Muribaculum[7–9] and these groups were also decreased in stressed mice (Fig. 1e–g). Bacterial diversity in the ileum was significantly reduced in mice exposed to psychological stress (Fig. 1h) but diversity change did not reach significance in the cecum or colon (Fig. 1i, j). Bacterial diversity in stressed mice clustered distinctly by PCA analysis in all three compartments, compared to samples from naive or starved controls (Fig. 1h–j). These data showed a clear treatment-based clustering of microbiota composition not only by PCA, but also by phylogenetic cluster analysis. Together, these results established a model to study the effects of acute psychological stress on the gut microbiome, showing that psychological stress produces a dramatic shift in the microbial population in the gut, with the most profound change being the expansion of ileal Enterobacteriaceae.

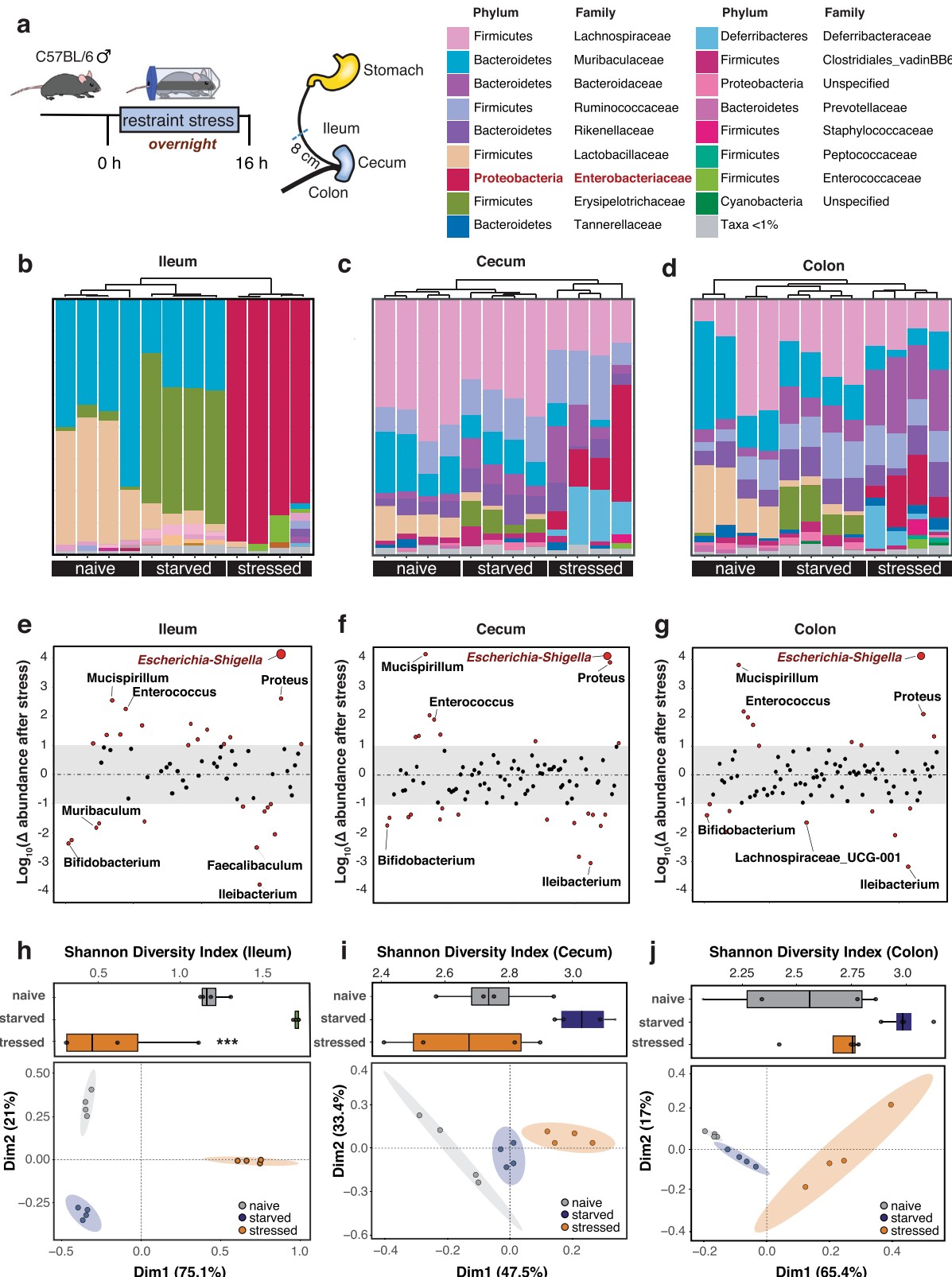

**Psychological stress impairs host control of CD-associated AIEC.** The dramatic expansion of ileal Enterobacteriaceae following acute psychological stress led us to question how CD-associated pathobionts would respond in this setting. To test this, mice were colonized with AIEC for 5 days and then subjected to overnight restraint stress or deprived of food and water overnight as a control (Fig. 2a). Six hours after release from stress, the fecal AIEC burden was significantly elevated by 4–5 orders of magnitude compared to the pre-stress burden. AIEC expansion was unrelated to the deprivation of food and water because AIEC levels remained unaltered in the starved control group (Fig. 2b). AIEC expansion was associated with the acute stress period because AIEC burdens normalized to pre-stress levels by 24 h after stress was relieved. To test the effects of periodicity on the

**Fig. 1 Psychological stress promotes the expansion of ileal Enterobacteriaceae. a** Schematic representation of the stress protocol and legend of bacterial phylum and family. Taxonomy plots of 16S rRNA sequencing of the ileal (**b**), cecal (**c**), or colonic (**d**), contents of naive ($n = 4$), starved (food and water deprived, $n = 4$), and restraint-stress ($n = 4$) exposed mice. Analysis of the proportional change in species abundance seen following restraint stress in the ileum (**e**), cecum (**f**), or colon (**g**). Change was calculated based on the relative proportional change of species abundance in stress mice, compared to the species abundance in the starved control. Shannon diversity index and principal component analysis of ileal (**h**), cecal (**i**), or colonic (**j**), 16S rRNA sequencing from naive ($n = 4$), starved ($n = 4$), and stress ($n = 4$) mice. PCA plots display a 95% confidence interval in the ileum, and 80% confidence interval in the cecum and colon. For Shannon diversity index, significance between starved and stressed mice $p = 0.0002$ was determined by one-way ANOVA. Box plot represents median and 25th and 75th percentiles—interquartile range; IQR—and whiskers extend to maximum and minimum values, adjusted for multiple comparisons ($*p \leq 0.05$; $**p \leq 0.01$; $***p \leq 0.001$).

stress-induced expansion and retraction of AIEC, we exposed AIEC-colonized mice to repeated weekly bouts of acute psychological stress for 1 month. Episodic exposure to stress produced a stereotyped expansion and contraction of AIEC and repeated exposures to stress progressively disabled host protection against AIEC (Fig. 2c). Whereas control mice deprived of food and water for one overnight period per week cleared AIEC between weeks 2 and 3, most mice exposed to acute psychological stress during the same diurnal period maintained high levels of AIEC beyond week 4 (Fig. 2c). These results indicated that exposure to acute psychological stress led to a marked expansion of AIEC and that repeated exposure to stress progressively disabled host control of AIEC. To determine where in the gut this AIEC expansion was occurring, we isolated the colon, cecum, and four equal segments of the small intestine following one round of overnight stress and enumerated AIEC in each compartment. AIEC significantly expanded along the entire length of the small and large bowel in stressed mice (Fig. 2d). Stress appeared to sensitize regions of the proximal small intestine to a greater extent because regions of the duodenum (segment 1) and jejunum (segment 2 & 3) that typically had no or low AIEC burden in the control mice were heavily colonized by AIEC in mice exposed to psychological stress (Fig. 2d). We sequenced 16S rRNA from the ileum of all groups of mice and again found a remarkable dominance of Enterobacteriaceae (Fig. 2e) and a significant reduction in community diversity (Fig. 2f). In non-stressed control mice, Enterobacteriaceae was detected in all mice, but accounted for <1% sequence abundance. Again, 16S rRNA profiles from the stressed group clustered distinctly from both the naive and starved control groups (Fig. 2f).

Given that both AIEC colonization and psychological stress have the capacity to exacerbate experimental colitis[43,47–49], we tested the effect of stress-associated morbidity on susceptibility to DSS-induced colitis in the presence or absence of AIEC. Stress alone induced a modest and transient weight loss that was recovered quickly prior to mice being switched from DSS to water. These mice lost only ~10% body weight and recovered fully within 2 weeks. However, mice infected with AIEC and exposed to stress had the most aggressive weight loss of ~20% (Fig. 2g). These mice failed to recover baseline weight even after 2 weeks following DSS treatment. Together, these results show that psychological stress impairs host control of CD-associated AIEC and contributes to increased susceptibility to DSS-induced colitis.

**Acute psychological stress impairs ileal barrier function and exposes mice to invasive microbes.** In the previous experiments, psychological stress worsened the severity of DSS-induced colitis in the presence of AIEC. Since AIEC colonization is enhanced during inflammation[13,14], we analyzed inflammatory cytokines and barrier function of the tissue in response to stress (Fig. 3a). In uninfected mice, stress alone induced only a modest increase in inflammatory cytokine mRNA. However, there was a dramatic increase for *TNF*, *IFN-γ*, *IL-23*, and *IL-17A* in the distal ileum of AIEC-infected mice exposed to acute psychological stress compared to controls (Fig. 3b). This inflammatory profile included a predominance of both $T_H17$ and $T_H1$ cytokines that we confirmed at the protein level in serum (Supplementary Fig. 2). We determined that TNF was dispensable for AIEC expansion because AIEC still bloomed in $TNF^{-/-}$ mice exposed to stress (Supplementary Fig. 3). Interestingly, the anti-inflammatory cytokine, IL-10, was also significantly upregulated following psychological stress in line with previous studies[50], although this also appeared to be dispensable as AIEC expansion still occurred following psychological stress in mice receiving an IL-10R blocking antibody (Supplementary Fig. 3).

Inflammatory cytokines can have a detrimental effect on gut permeability by downregulating genes involved in barrier function[51,52]. We profiled the expression of genes associated with barrier function and measured ileal permeability in Ussing chambers. In mice exposed to stress, there was a generalized reduction in gene expression for the mucins *Muc2*, *Muc3*, and *Muc4* and downregulation of the tight junction genes claudin 2 and occludin (Fig. 3c, Supplementary Fig. 4). In contrast, AIEC-colonized stressed mice exhibited a dramatic and significant increase in the transcription factor, GATA-4 (Fig. 3c), which plays a central role in epithelial repair in the small intestine, strongly indicative of stress-induced epithelial damage[53]. To directly measure ileal permeability, we mounted explanted ileal tissues in Ussing chambers and measured paracellular passage of radiolabeled chromium. Tissues from both uncolonized and AIEC-colonized mice showed significantly increased paracellular flux when the mice were exposed to stress (Fig. 3d), indicating that stress contributes to functional barrier disruption in the ileum and that the presence of AIEC exacerbates this disruption.

Given the enhanced inflammation and increased intestinal permeability associated with stress, we tested the functional consequences of these barrier defects on containment of intestinal bacteria. Significantly elevated levels of LPS (Fig. 3e) and bacterial dissemination to the liver (Fig. 3f) were found in AIEC-colonized mice exposed to psychological stress. Whereas 60% of control mice had undetectable levels of bacteria in their livers, only 25% of mice exposed to stress had sterile livers. TLR-4 activation is linked to increased intestinal permeability[54,55]. To further confirm this finding and evaluate the immunological impact of increased bacterial dissemination under stress, we measured *TLR-4* expression in the ileum by RT-qPCR, which was ~8–16-fold higher in mice exposed to stress compared to unstressed controls (Fig. 3g). The activation of several downstream targets of TLR-4 signaling confirmed the activity of this pathway, including a ~64-fold increase in *IL-6* mRNA levels in the ileum of mice exposed to stress compared to controls (Fig. 3h). These findings were validated and confirmed in serum samples at the protein level (Supplementary Fig. 2). In keeping with our model that AIEC exacerbates stress-induced mucosal defects, naive, uninfected mice exposed to stress had lower bacterial dissemination and reduced upregulation of TLR-4 and IL-6 (Supplementary Fig. 4). The upregulation of the neutrophil chemokine, KC, in the serum of infected mice exposed to stress was consistent with systemic bacterial leakage

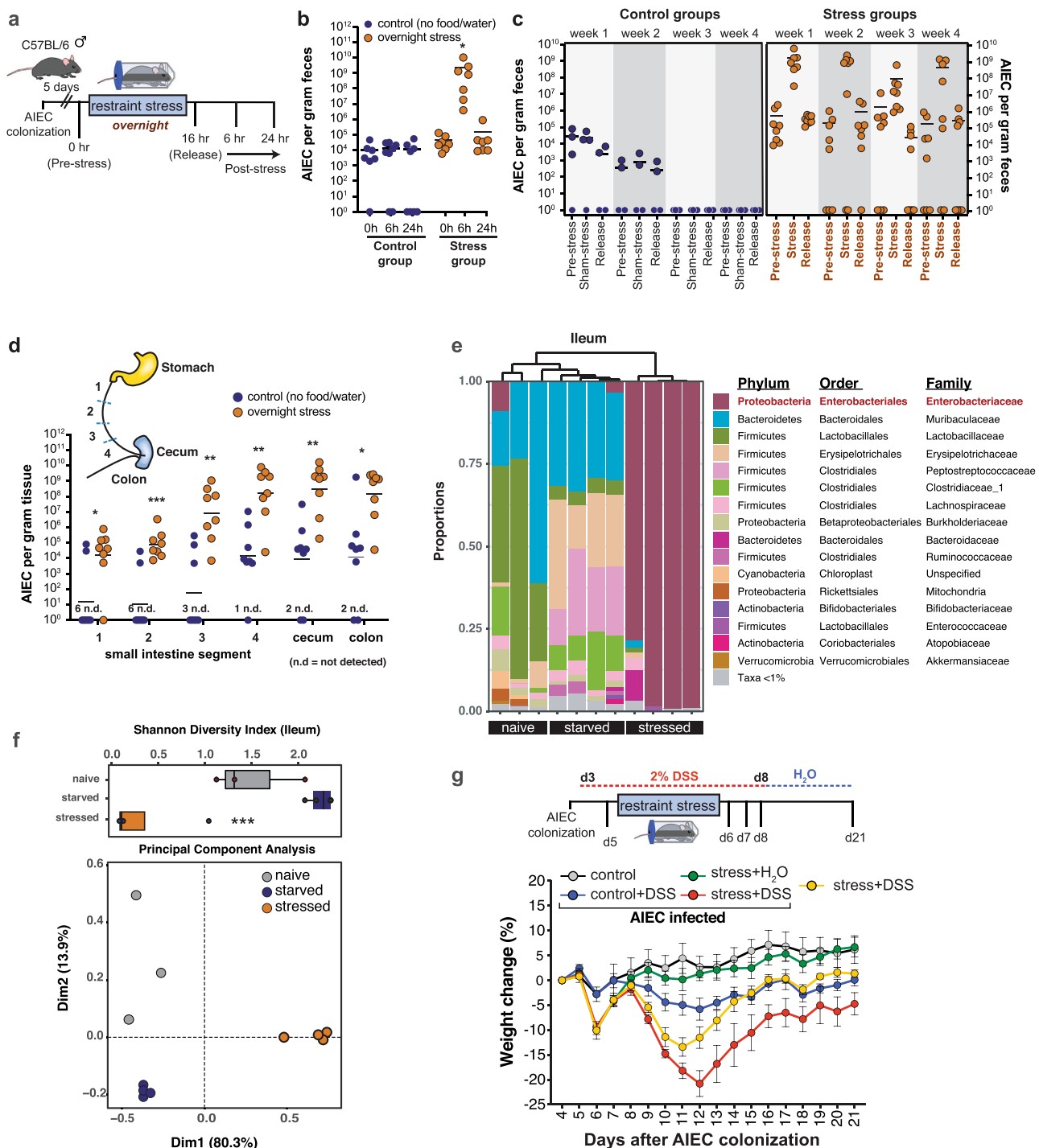

**Fig. 2 Psychological stress impairs host control of CD-associated AIEC. a** Schematic representation of infection and stress protocol. **b** AIEC fecal burdens collected from control ($n = 8$) and stress ($n = 7$) exposed mice at the time of stress (or overnight starvation), and at 6 and 24 h post-stress. Stress-exposed mice were compared to matched controls by a two-way ANOVA, not corrected for multiple comparisons, $p = 0.0194$. **c** Mice were colonized with AIEC and subjected to either overnight stress ($n = 8$) or starvation ($n = 4$) for 16 h weekly for 4 weeks. Fecal samples were collected at the time of stress (or overnight starvation), and at 6 h and 24 h post-stress. **d** AIEC tissue burdens collected from the length of the intestinal tract from control ($n = 8$) and stress ($n = 8$) mice. The small intestine was divided into four 8 cm segments in which segment 4 is adjacent to the cecum. Stress-exposed mice were compared to match controls by a two-tailed Mann–Whitney test, $p$ values = 0.022067, 0.000932, 0.001088, 0.001865, 0.001088, 0.020202. **e** Taxonomy plot of 16S rRNA sequencing of ileum in AIEC-colonized naive ($n = 3$), starved (food and water depleted, $n = 4$), and restraint-stress ($n = 4$) exposed mice. **f** Shannon diversity index and principal component analysis of ileal 16S rRNA sequencing from AIEC-colonized naive ($n = 3$), starved ($n = 4$) and stress ($n = 4$) mice. For Shannon Diversity Index, significance between starved and stressed mice, $p = 0.0004$ was determined by one-way ANOVA. Box plot represents median and 25th and 75th percentiles—interquartile range; IQR—and whiskers extend to maximum and minimum values, adjusted for multiple comparisons. **g** Schematic representation of DSS treatment schedule. The graph depicts weight change from the time of DSS initiation (all groups are $n = 4$) (*$p \leq 0.05$; **$p \leq 0.01$; ***$p \leq 0.001$). Error bars represent SEM and the line in CFU graphs indicates the geometric mean of the group.

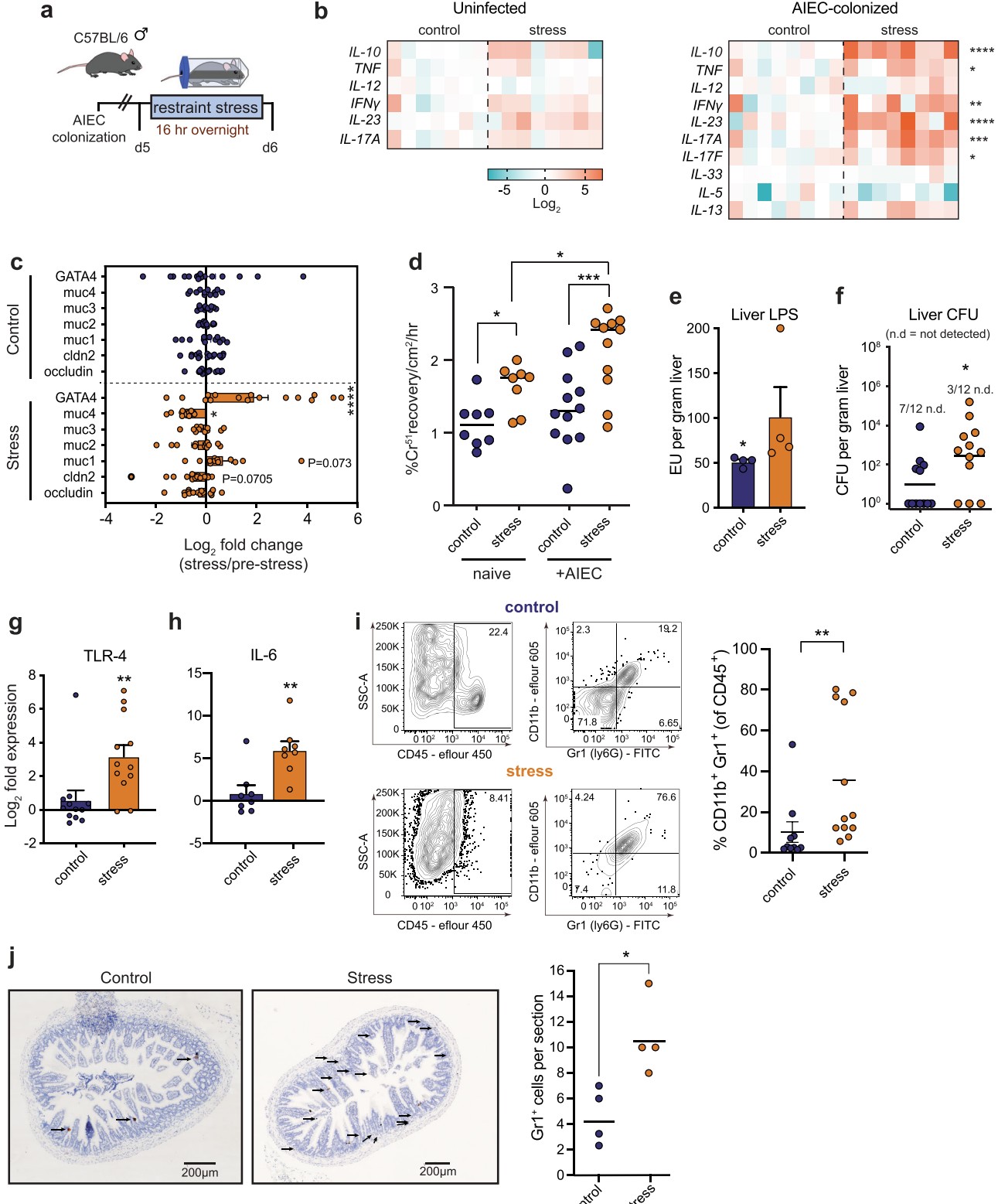

(Supplementary Fig. 2). Indeed, flow cytometry revealed that stress induced a marked recruitment of $CD45^+CD11b^+Gr1^+$ immune cells to the ileum, a cell population consistent with neutrophils (Fig. 3i). An increase in $GR1^+$ cells in the tissue was confirmed by immunohistochemistry (Fig. 3j). Together, these findings indicated that stress induces a profound impairment of intestinal barrier function with an attendant inflammatory response that is exacerbated in the presence of AIEC.

**AIEC derives a competitive advantage from stress-induced nutritional immunity.** Breach of the mucosal epithelium induces nutritional immunity through the release of host molecules that sequester metals, such as iron, as means of limiting bacterial outgrowth[56–58]. To probe the effect of stress-induced barrier breach on nutritional immunity, we measured by RT-qPCR the expression of host genes associated with nutritional immunity and metal restriction. Following exposure to psychological stress

**Fig. 3 Exposure to acute psychological stress impairs ileal barrier function and exposes mice to invasive microbes. a** Schematic representation of infection and stress protocol. **b** RT-qPCR analysis of cytokine expression of the ileum of naive starved ($n = 7$), naive stressed ($n = 8$), AIEC-colonized starved ($n = 8$), AIEC-colonized stressed ($n = 8$) samples. Significance was determined by a two-way ANOVA, not corrected for multiple comparisons, IL-10, $p < 0.0001$; TNF, $p = 0.031$; IFN-γ, $p = 0.0032$; IL-23, $p < 0.0001$; IL-17A, $p = 0.0004$; IL-17F, $p = 0.0383$. **c** RT-qPCR analysis of RNA expression for genes associated with barrier function in the ileum of AIEC-colonized starved and stressed mice. Significance was determined by a two-way ANOVA, not corrected for multiple comparisons ($n =$ occludin (c16, s16); cldn2 (c16, s16); muc1 (c12, s12); muc2 (c12, s12); muc3 (c12, s12); muc4 (c12, s12); GATA4 (c16, s16)). GATA4, $p < 0.0001$; Muc4, $p = 0.0294$. **d** Paracellular permeability determined by chromium passage through a 1 cm ileal segment collected from naive ($n = 8$) or AIEC-colonized ($n = 12$) mice subjected to starvation or restraint stress. Significance was determined by a two-way ANOVA, not corrected for multiple comparisons, control (naive):stress (naive), $p = 0.0362$; control (infected):stress (infected), $p = 0.0003$; stress (naive):stress (infected), $p = 0.0346$. **e** Liver LPS concentration, represented as Endotoxin Units per gram tissue, was quantified within AIEC-colonized starved ($n = 4$) and stressed ($n = 4$) mice. Significance was determined by a two-tailed Mann–Whitney, $p = 0.0286$. **f** Bacterial burdens in the liver were enumerated in AIEC-colonized starved ($n = 12$) or stressed ($n = 12$) mice. Significance was determined by a two-tailed Mann–Whitney, $p = 0.0240$. **g** RT-qPCR analysis of TLR-4 RNA expression in the ileum of AIEC-colonized starved ($n = 12$) or stressed ($n = 12$) mice. Significance was determined by a two-tailed Mann–Whitney, $p = 0.0056$. **h** RT-qPCR analysis of IL-6 RNA expression in the ileum of AIEC-colonized starved ($n = 8$) or stressed ($n = 8$) mice. Significance was determined by a two-tailed Mann–Whitney, $p = 0.0070$. **i** Representative FACS plots of CD45$^+$CD11b$^+$GR1$^+$ cells. Percentage of CD11b$^+$Gr1$^+$ cells from the total population of CD45$^+$ cells isolated from ileal lamina propria cells collected from starved ($n = 10$) and stressed ($n = 12$) mice as determined by flow cytometry. Significance was determined by a two-tailed Mann–Whitney, $p = 0.0040$. **j** Number of GR1$^+$ cells as determined by immunohistochemistry. Significance was determined by a two-tailed Mann–Whitney test, $p = 0.0286$. (*$p \leq 0.05$; **$p \leq 0.01$; ***$p \leq 0.001$; ****$p \leq 0.0001$). Error bars represent SEM and the line in CFU graphs indicates the geometric mean of the group.

(Fig. 4a), the genes *HMOX−1* (heme oxygenase-1) and *Nramp1* (natural resistance-associated macrophage protein-1) were significantly upregulated in the ileum (Fig. 4b), indicating that stress was associated with an induction of nutritional immunity. Lcn2 and the S100a8 and S100a9 subunits of calprotectin are well-known metal-binding proteins that are upregulated in the inflamed intestine[23,27,59]. In addition to *HMOX-1* and *Nramp1*, mice exposed to psychological stress had an increase in ileal mRNA for *S100a8*, *S100a9*, and *Lcn2* (Fig. 4b), which we confirmed at the protein level by ELISA in AIEC colonized (Fig. 4c, d) and in naive stress-exposed mice (Supplementary Fig. 4). To follow the kinetics of this host response, we measured Lcn2 and S100a9 levels after 4 or 16 h of stress, and after 6 or 12 h following release from overnight stress. Enhanced expression of *Lcn2* and *S100a9* was seen in both the ileum and liver after 4 h of stress (Fig. 4e). RT-qPCR analysis revealed that enhanced Lcn2 expression was maintained over the course of stress and remained elevated even 6 h following stress release and recovery (Fig. 4f). A corresponding increase in ileal AIEC burdens accompanied elevated levels of Lcn2 during stress and followed a similar dynamic (Fig. 4g), with a significant positive correlation between AIEC and Lcn2 expression in the ileum (Fig. 4h, Supplementary Fig. 5). A similar correlation with AIEC burden was evident for S100a8 and Lcn2 in the cecum (Fig. 4h).

Given the correlation between Lcn2 expression and AIEC expansion following stress, we sought to uncover the mechanism by which AIEC derives a competitive advantage over commensal members of the microbiome. Unlike most commensal *E. coli*, AIEC encodes the *iroBCDE iroN* gene cluster involved in the biosynthesis and transport of salmochelin, a glycosylated form of enterobactin that is resistant to Lcn2 sequestration[60]. Under conditions of high Lcn2, as we observed during psychological stress, bacteria expressing salmochelin would have a competitive advantage over non-salmochelin-expressing bacteria. Indeed, whereas a Δ*iroB* mutant competed equally with wild-type AIEC under non-stressed conditions, this mutant was significantly outcompeted throughout the gut in mice exposed to overnight restraint stress (Fig. 4i), confirming that stress creates a selective host environment for microbes that are tolerant to nutritional immunity. Together, these data show that AIEC exploits nutritional immunity induced following stress, in part, through the expression of iron-scavenging siderophores that are resistant to host sequestration.

**Psychological stress induces glucocorticoid receptor-dependent apoptosis in CD45$^+$CD90$^+$ cells.** Despite robust recruitment of CD11b$^+$Gr1$^+$ immune cells and potent induction of nutritional immunity, the AIEC expansion observed during stress suggested other impairments to protective host function. We first confirmed that stress was necessary for AIEC expansion by exogenously inducing Lcn2 expression by intraperitoneal delivery of LPS in the absence of stress. Indeed, although LPS delivery alone significantly induced Lcn2 expression in the ileum (Fig. 5a), this treatment was insufficient to cause AIEC expansion (Fig. 5b), strongly suggesting that stress was having a multidimensional effect on host immunity required for AIEC expansion. Gut lymphocytes are key players in homeostasis and host defense. Immunophenotyping of ileal lamina propria cells revealed a marked reduction in CD90$^+$ T cells and CD90$^+$ ILCs following stress (Fig. 5c), consistent with previous observations of lymphocytic attrition following stress[51,61]. To better understand how stress signaling was leading to a loss of CD90$^+$ lymphocytes, we established readouts for the glucocorticoid corticosterone during stress (Fig. 5d) and the expression of the glucocorticoid receptor on the CD45$^+$CD90$^+$ cell population, which indicated significantly elevated levels of corticosterone during stress and that most CD45$^+$CD90$^+$ cells expressed glucocorticoid receptor (Fig. 5e). Next, we evaluated the CD90$^+$ lymphocyte population shortly after stress induction to identify early markers of lymphocyte attrition. Through empirical determination, 8 h into stress was a time point that preceded the onset of lymphocyte loss. Interestingly, although the frequency of CD90$^+$ lymphocytes was similar to control mice at this early time point (Fig. 5f), CD90$^+$ cells in stress-exposed mice already had a significant upregulation of the early marker of apoptosis, Annexin V. The frequency of Annexin V-positive cells was maintained at baseline control levels by pre-treating mice with the glucocorticoid receptor antagonist, RU486 (Fig. 5g). Indeed, both the frequency (Fig. 5h) and absolute number of CD90$^+$ lymphocytes (Fig. 5i) was significantly reduced following stress and this cell population was rescued by glucocorticoid receptor blockade. These results established a direct relationship between stress-signaling and the loss of CD45$^+$CD90$^+$ cells in the gut.

The CD45$^+$CD90$^+$ cell population comprises both T cells and ILCs. To determine if a specific subset of CD45$^+$CD90$^+$ cells was being preferentially depleted following stress, we immunophenotyped this population from the ileum in control and stressed

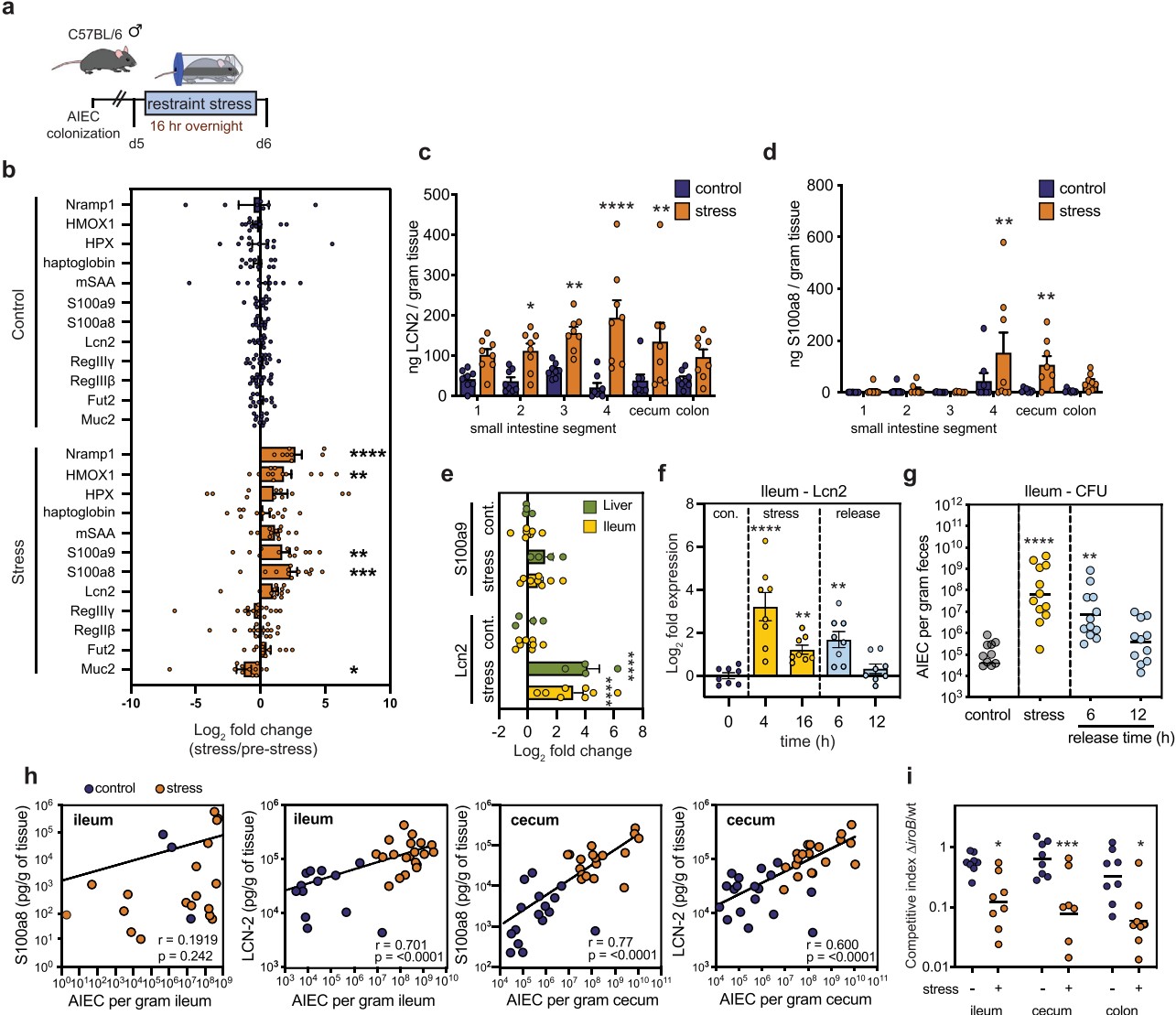

**Fig. 4 Stress-induces nutritional immunity within the gut provides a competitive advantage to AIEC. a** Schematic representation of infection and stress protocol. **b** RT-qPCR analysis from ileal samples of AIEC-colonized starved ($n = 12-16$) or stressed ($n = 12-16$) mice. Significance was determined by two-way ANOVA, not corrected for multiple comparisons ($n$ = Muc2 (c-12, s-12); Fut2 (c-12, s-12); Reg3β (c-16, s-16); Reg3γ (c-16, s-16); Lcn2 (c-12, s-12); S100a8 (c-12, s-12); S100a9 (c-12, s-12); mSAA (c-12, s-12); haptoglobin (c-11, s12); HPX (c-12, s-11); HMOX1 (c-11. s-12); nRAMP1 (c-7, s-8)); Muc2 $p = 0.0452$; S100a8 $p = 0.0003$; S100a9 $p = 0.0092$; HMOX1 $p = 0.0017$; NRAMP1 $p < 0.0001$. **c** Quantification of Lcn2 by ELISA along the length of the intestinal tract in AIEC-colonized starved ($n = 8$) and stressed ($n = 8$) mice. The small intestine was divided into four 8 cm segments in which segment 4 was proximal to the cecum. Significance was determined by two-way ANOVA, not corrected for multiple comparisons, Seg 1 $p = 0.0551$; Seg 2 $p = 0.0164$; Seg 3 $p = 0.0033$; Seg 4 $p < 0.0001$; Cecum $p = 0.0023$; Colon $p = 0.0754$. **d** Quantification of the calprotectin subunit S100a8 along the length of the intestinal tract in AIEC-colonized starved ($n = 8$) and stressed ($n = 8$) mice. Significance was determined by two-way ANOVA, not corrected for multiple comparisons, Seg 4 $p = 0.0042$, Cecum $p = 0.0088$. **e** RT-qPCR analysis of *S100a9* and *Lcn2* RNA expression in the liver and ileum of AIEC-colonized mice following 4 h of starvation (ileum $n = 8$, liver $n = 4$) or stress (ileum $n = 8$, liver $n = 4$). Significance to control was determined by one-way ANOVA, not corrected for multiple comparisons, Lcn2 control:stress, ileum, $p < 0.0001$; Lcn2 control:stress, liver, $p < 0.0001$. **f** RT-qPCR analysis of Lcn2 in the ileum. Samples were collected from AIEC-colonized mice at baseline as a control, 4 and 16 h of overnight stress, and at 6 and 12 h following restraint release (for all time points $n = 8$). Significance to control was determined by one-way ANOVA, not corrected for multiple comparisons, 4 h $p < 0.0001$; 16 h, $p = 0.0070$; 6 h, $p = 0.0017$. **g** AIEC fecal burdens collected from mice at baseline as a control ($n = 11$), 16 h of stress ($n = 12$), and at 6 h ($n = 12$) and 12 h ($n = 11$) following restraint release and recovery. Significance to control was determined by one-way ANOVA, not corrected for multiple comparisons, stress, $p < 0.0001$; 6 h release, $p = 0.0016$. **h** A two-tailed Spearman correlation of either calprotectin subunit S100a8 concentration or Lcn2 concentration and AIEC fecal burdens in the ileum and cecum ($n = 40$), ileum, $p < 0.0001$; cecum, $p < 0.0001$. **i** Competitive infection of wild-type AIEC and Δ*iroB* in starved ($n = 8$) and stressed ($n = 8$) mice. Mice were infected with a 1:1 ratio of wild type:Δ*iroB* and were subjected to overnight starvation or stress 2 days later. Following stress or starvation, ileum, cecum, and colon tissues were collected and the ratio of wild type:Δ*iroB* determined by selective plating. Significance was determined by a two-way ANOVA, not corrected for multiple comparisons, ileum, $p = 0.0148$; cecum, $p = 0.0005$; colon, $p = 0.0201$ (*$p ≤ 0.05$; **$p ≤ 0.01$; ***$p ≤ 0.001$). Error bars represent SEM and the line in CFU graphs indicates the geometric mean of the group.

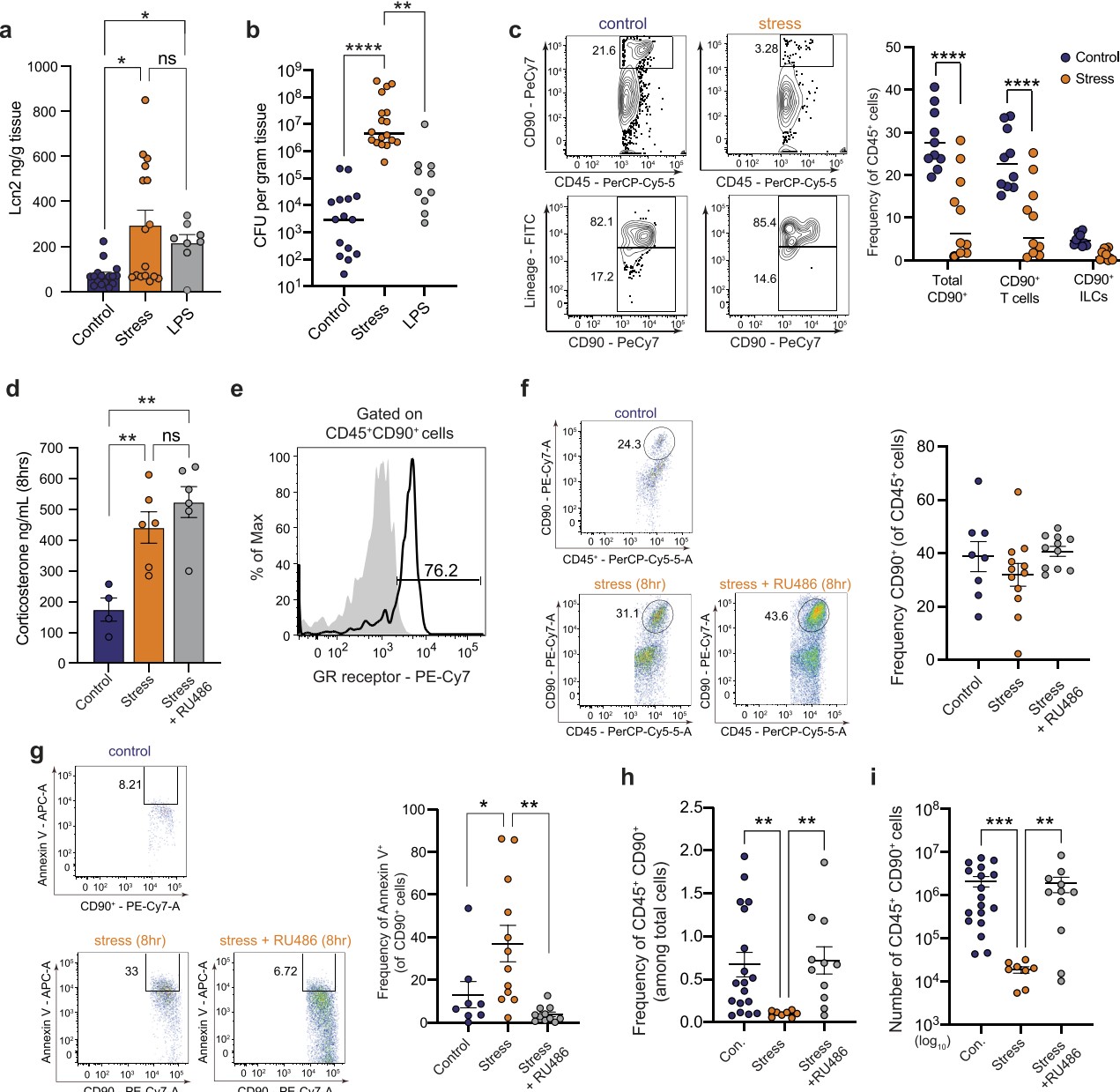

**Fig. 5 Psychological stress induces attrition of CD90+ cells, resulting in defects in the IL-22 pathway. a** Quantification of ileal Lcn2 by ELISA in control ($n = 16$), stress ($n = 16$), and LPS treated ($n = 8$) mice. Significance was determined by a one-way ANOVA, not corrected for multiple comparisons, control:stress, $p = 0.0141$, control:LPS, $p = 0.0466$. **b** Ileal AIEC tissue burdens from control ($n = 16$), stress ($n = 18$), and LPS treated ($n = 10$) mice control:stress, $p < 0.0001$, control:LPS, $p = 0.0079$. **c** Representative flow plots of CD45+CD90+/− cells and lineage staining on CD45+CD90+. Percentage of CD90+ T cells and CD90+ lineage− ILCs of total CD45+ cells isolated from ileal lamina propria cells collected from starved ($n = 10$) and stressed ($n = 10$) mice as determined by flow cytometry. Significance was determined by two-way ANOVA, not corrected for multiple comparisons, $p < 0.0001$, $p < 0.0001$. **d** Quantification of corticosterone in the serum as determined by ELISA in control ($n = 4$), stress ($n = 6$), and RU486 treated stress ($n = 6$) mice. Significance was determined by one-way ANOVA, not corrected for multiple comparisons, control:stress, $p = 0.0087$; control:stress + RU486, $p = 0.0011$. **e** Representative flow plot of percentage of CD45+CD90+GR+ cells. **f** Frequency of CD45+CD90+ cells following 8 h of restraint stress in control ($n = 8$), stress ($n = 12$), and RU486 treated stress ($n = 11$) mice. **g** Frequency of CD45+CD90+AnnexinV+ cells following 8 h of restraint in control ($n = 8$), stress ($n = 12$), and RU486 treated stress ($n = 11$). Significance was determined by a one-way ANOVA, not corrected for multiple comparisons, control:stress, $p = 0.0323$, stress:stress + ru486 $p = 0.0018$. **h** Frequency of CD45+CD90+ cells following overnight restraint in control ($n = 18$), stress ($n = 8$), and RU486 treated stress ($n = 11$). Significance was determined by one-way ANOVA, not corrected for multiple comparisons, control:stress, $p = 0.0079$, stress:stress + ru486 $p = 0.0043$. **i** Absolute number of CD45+CD90+ cells following overnight restraint in control ($n = 18$), stress ($n = 8$), and RU486 treated stress ($n = 11$) mice. Significance was determined by two-way ANOVA, not corrected for multiple comparisons, control:stress, $p = 0.0005$, stress:stress + ru486 $p = 0.0060$. (*$p \le 0.05$; **$p \le 0.01$; ***$p \le 0.001$; ****$p \le 0.0001$). Error bars represent SEM and the line in CFU graphs indicates the geometric mean of the group.

animals. These experiments showed that although stress significantly reduced the overall population numbers of CD45[+]CD90[+] cells, as we demonstrated earlier, the frequency of specific cell subsets was similar between control and stressed groups. Both CD90[+]TCRβ[−] cells and CD90[+]TCRβ[+] cell frequencies were similar between control and stress groups in the small intestine, as were the various TCRβ[+] T cell populations (CD4[+], CD8[+], or CD4[+]CD8[+] T cells) (Supplementary Fig. 8). We also confirmed overall reductions in the absolute numbers of T$_H$17 and ILC3 cell subsets, which are major regulators of inflammation and mucosal protection in the gut, but the frequency of these populations was similar between control and stressed mice (Supplementary Fig. 8). Together, these data showed that stress induces an indiscriminate contraction of the CD45[+]CD90[+] cell population and not a specific subpopulation.

**Psychological stress impairs IL-22-dependent host immunity.** In the gut, CD45[+]CD90[+] cells express IL-22, a critical cytokine involved in host defense at the mucosal surface[24]. Given the depletion of CD45[+]CD90[+] cells following stress, we reasoned that stress might be blunting protective IL-22 responses in the presence of invasive bacteria like AIEC. To determine whether depletion of CD45[+]CD90[+] cells influenced IL-22 production, we cultured ileal explants from control mice, stressed mice, and stressed mice treated with RU486 and compared their response ex vivo to stimulation with IL-23, a treatment expected to induce the expression of IL-22. Whereas ileal explants from control mice mounted a stereotyped induction of IL-22 following IL-23 stimulation, explants from stressed mice did not (Fig. 6a). IL-22 induction was restored in ileal explants harvested from stressed mice treated with RU486 (Fig. 6a), providing a direct link between stress signaling and IL-22 depletion. Given the dramatic loss of CD45[+]CD90[+] lymphocytes following stress, these data strongly suggest that stress impairs the IL-22 axis through a reduction in T$_H$17 and ILC3 cells, as these CD45[+]CD90[+] cell subsets are the main producers of IL-22 in the gut.

Our previous data indicated that psychological stress induced a state of nutritional immunity and immune cell depletion. Since delivery of LPS alone to non-stressed mice elicited Lcn2 expression but not AIEC expansion, these data suggested that immune cell attrition was the second condition required for AIEC expansion in stressed mice. If this hypothesis was correct, then immune cell depletion in the absence of stress would also fail to elicit AIEC expansion because the nutritional immunity pathway would not be engaged. To test this hypothesis, we first depleted either CD90[+] cells or IL-22 in AIEC colonized mice and measured AIEC burden in the ileum compared to IgG control-treated mice. In the absence of stress, neither CD90[+] cell depletion or IL-22 depletion alone led to AIEC expansion in the gut, consistent with the notion that stress is required to invoke a secondary pathway of nutritional immunity (Fig. 6b). Even treating unstressed mice with anti-IL-22 neutralizing antibody over a prolonged course of 11 days did not result in perturbation of AIEC levels in the feces (Supplementary Fig. 6). Next, we subjected AIEC-colonized mice to psychological stress and blocked glucocorticoid signaling with RU486, which we showed previously protects the CD90[+] cell population from depletion (Fig. 5i), but still elicits Lcn2 release (not shown). Under these conditions, RU486 treatment prevented the stress-induced expansion of AIEC (Fig. 6c), and consistent with our model, co-administration of either anti-CD90 antibody or anti-IL-22 antibody phenocopied the impact of stress, leading to AIEC expansion (Fig. 6c). As a final validation of our hypothesis that both nutritional immunity and immune cell depletion are pathways downstream from stress leading to AIEC expansion, we reconstituted both pathways in the absence of stress using LPS

administration and CD90[+] cell depletion, which would be expected to facilitate AIEC outgrowth according to our model. Indeed, in mice where nutritional immunity was activated via LPS administration and CD90[+] cells were concurrently depleted, AIEC expanded to levels similar to that seen during stress exposure (Fig. 6d). Together, these data confirmed that stress-induced AIEC expansion requires both induction of nutritional immunity and glucocorticoid-mediated depletion of IL-22-producing immune cells.

**IL-22 delivery corrects stress-induced impairments to mucosal immunity and prevents intestinal dysbiosis.** Having determined that psychological stress impairs IL-22-driven protective mucosal immunity against Crohn's disease-associated pathobionts, we reasoned that reconstituting IL-22 during stress should correct this defect and restore mucosal defenses. We treated AIEC-colonized mice with IL-22-Fc or an equivalent amount of isotype control protein, exposed the mice to stress, and measured AIEC burdens. Whereas control-treated mice showed an expected increase in AIEC burdens following stress, mice treated with IL-22-Fc had a significantly blunted outgrowth of AIEC in the ileum (Fig. 7a), as well as the cecum and colon (Supplementary Fig. 7). Signal transducer and activator of transcription (STAT)-3 links IL-22 signaling to epithelial host defense in the gut[62–64]. Given our results that implicated insufficient IL-22 levels following stress, we stained isolated gut epithelial cells for phosphorylated STAT-3 and measured these levels by flow cytometry. In agreement with an insufficient level of IL-22 during stress, a significantly lower proportion of epithelial cells were positive for phospho-STAT-3 after stress compared to unstressed controls (Fig. 7b). In contrast, IL-22-Fc-treatment partially restored the phospho-STAT-3-positive epithelial cell population in the ileum following psychological stress (Fig. 7b).

To understand how IL-22 might be normalizing the ileal microenvironment and preventing Enterobacteriaceae expansion, we subjected bulk ileal tissue to RNA sequencing. Our analysis showed that stress induced a distinct ileal transcriptome that was largely normalized by IL-22-Fc treatment (Fig. 7c). Similar to the cytokine changes seen in earlier experiments, psychological stress induced a generalized upregulation of pro-inflammatory cytokine transcript expression and an overall reduction in several innate antimicrobial pathways (Fig. 7c, d). This dataset confirmed that stress affects both IL-22-dependent and IL-22-independent antimicrobial pathways (Fig. 7d). Specifically, stress reduced the expression of several antimicrobial alpha defensins independent of IL-22, while the expression of beta defensins was modestly induced following stress (Fig. 7d). Moreover, mice exposed to stress had decreased expression of genes associated with the production of reactive oxygen and nitrogen species, as well as genes in the phospholipase A2 family that have broad antimicrobial activity. These deficiencies were correctable by IL-22-Fc administration. The Reg3 family of antimicrobial lectins was highly induced by IL-22-Fc treatment, suggesting that IL-22-inducible AMPs can compensate for the loss of other innate antimicrobial pathways during stress (Fig. 7d). In line with this, we confirmed by RT-qPCR a generalized normalization by IL-22-Fc of pro-inflammatory cytokines that reached significance for IL-23, IL-17A, IFN-γ, and IL-6 (Fig. 7e). Furthermore, RT-qPCR confirmed a dichotomy of IL-22-dependent antimicrobial pathways affected by stress characterized by a generalized suppression of AMPs while preserving the activation of nutritional immunity (Fig. 7f), consistent with our earlier results. Exogenous IL-22-Fc administration significantly increased IL-22-inducible AMP expression compared to stressed controls throughout the intestinal tract (Fig. 7f, Supplementary Fig. 7).

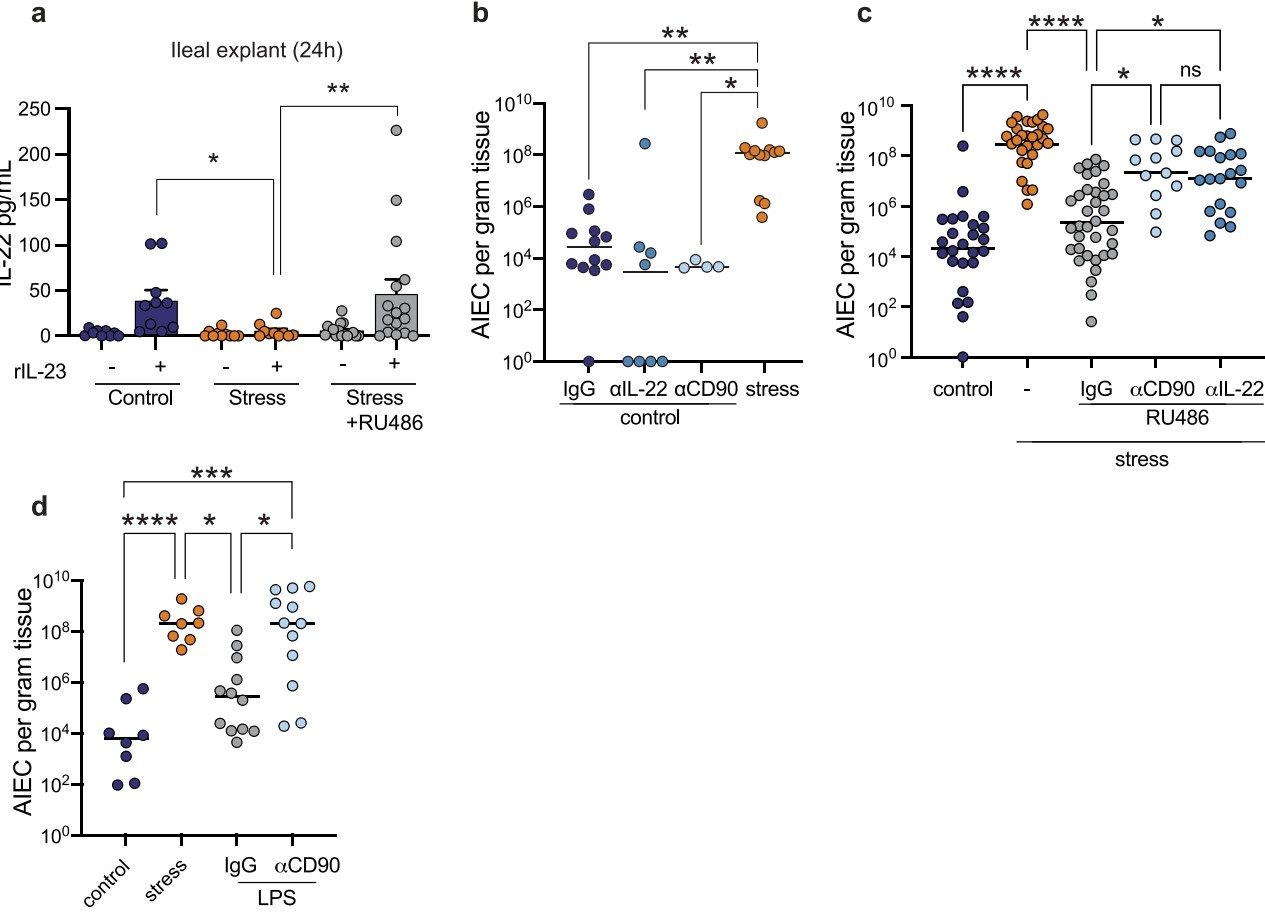

**Fig. 6 Combinatorial effects of nutritional immunity and immune depletion are responsible for intestinal dysbiosis. a** Quantification of IL-22 in the supernatants of ileal explants from starved ($n = 10$), stressed ($n = 10$), RU486 treated stress ($n = 16$) mice stimulated with 20 ng/mL rIL-23 or media control for 24 h. Significance was determined by one-way ANOVA, not corrected for multiple comparisons, control:stress $p = 0.0285$, stress:stress RU486, $p = 0.0033$. **b** Ileal AIEC tissue burdens from control ($n = 12$), αIL-22 ($n = 8$), αCD90 treated ($n = 4$), and stress ($n = 12$) mice. Significance was determined by one-way ANOVA, not corrected for multiple comparisons, control:stress, $p = 0.0063$; control αIL-22: stress, $p = 0.0013$; control αCD90:stress, $p = 0.0106$. **c** Ileal AIEC tissue burdens from control ($n = 19$), stress ($n = 22$), RU486 treated stress (n = 25), RU486 and αCD90 treated stress ($n = 12$), and RU486 and αIL-22 treated stress ($n = 13$). Significance was determined by one-way ANOVA, not corrected for multiple comparisons, control:stress, $p < 0.0001$; stress:stress + RU486 IgG, $p < 0.0001$; stress + RU486 IgG:stress + RU486 αCD90 $p = 0.0284$; stress + RU486 IgG:stress + RU486 αIL-22, $p = 0.0301$. **d** Ileal AIEC tissue burden in control ($n = 24$), stress ($n = 18$), LPS treated ($n = 22$), and LPS and αCD90 treated ($n = 12$) mice. Significance was determined by one-way ANOVA, not corrected for multiple comparisons, control:stress, $p = 0.0006$; control:LPS + αCD90, $p = 0.0003$; stress:LPS, $p = 0.0399$; LPS:LPS + αCD90, $p = 0.0357$. (*$p ≤ 0.05$; **$p ≤ 0.01$; ***$p ≤ 0.001$; ****$p ≤ 0.0001$). Error bars represent SEM and the line in CFU graphs indicates the geometric mean of the group.

To test whether IL-22 could correct the stress-induced augmentation of DSS-induced weight loss seen previously, we pre-treated DSS exposed mice with IL-22-Fc prior to stress and then every other day after release from stress (Fig. 7g). As with our previous data, stress significantly exaggerated DSS-induced weight loss relative to unstressed mice receiving DSS. However, mice treated with IL-22-Fc lost significantly less weight, indicative of reduced DSS-induced illness. Finally, to test the impact of IL-22 correction on stress-induced ileal dysbiosis, we profiled the ileal microbiome from control mice, and mice exposed to stress with or without IL-22 treatment. IL-22-Fc treatment of mice prior to stress largely prevented the profound expansion of ileal Enterobacteriaceae that was reproducibly seen in mice exposed to stress alone (Fig. 7h) and almost completely prevented this expansion in the cecum and colon (Supplementary Fig. 7). Altogether, these findings indicate that psychological stress impairs IL-22-driven protective mucosal immunity against CD-associated pathobionts and that this defect is correctable with exogenous IL-22 treatment.

## Discussion

CD patients experiencing psychological stress are more likely to relapse and have increased disease activity, yet the mechanisms underlying this remain obscure[33,65–67]. Likewise, despite a higher frequency of AIEC colonization in CD patients, the host factors that influence the fluctuation of AIEC load in patients are not understood[68]. Current evidence indicates that AIEC is a patho-biont in the human gut and takes advantage of inflammatory niches derived from host insults, such as antibiotic use[69], secondary enteric infections[20], western diet[70], and/or genetic predisposition[71]. In this study, we determined that psychological stress is associated with impaired IL-22-dependent mucosal antimicrobial defenses that result in profound small intestinal dysbiosis dominated by AIEC expansion.

Previous studies have demonstrated a link between intestinal dysbiosis and inflammation in mice[47,72,73] and in IBD patients[7]. In our study, exposure to acute psychological stress resulted in a profound dysbiosis dominated by outgrowth of Enterobacter-iaceae family members and genera commonly found to be

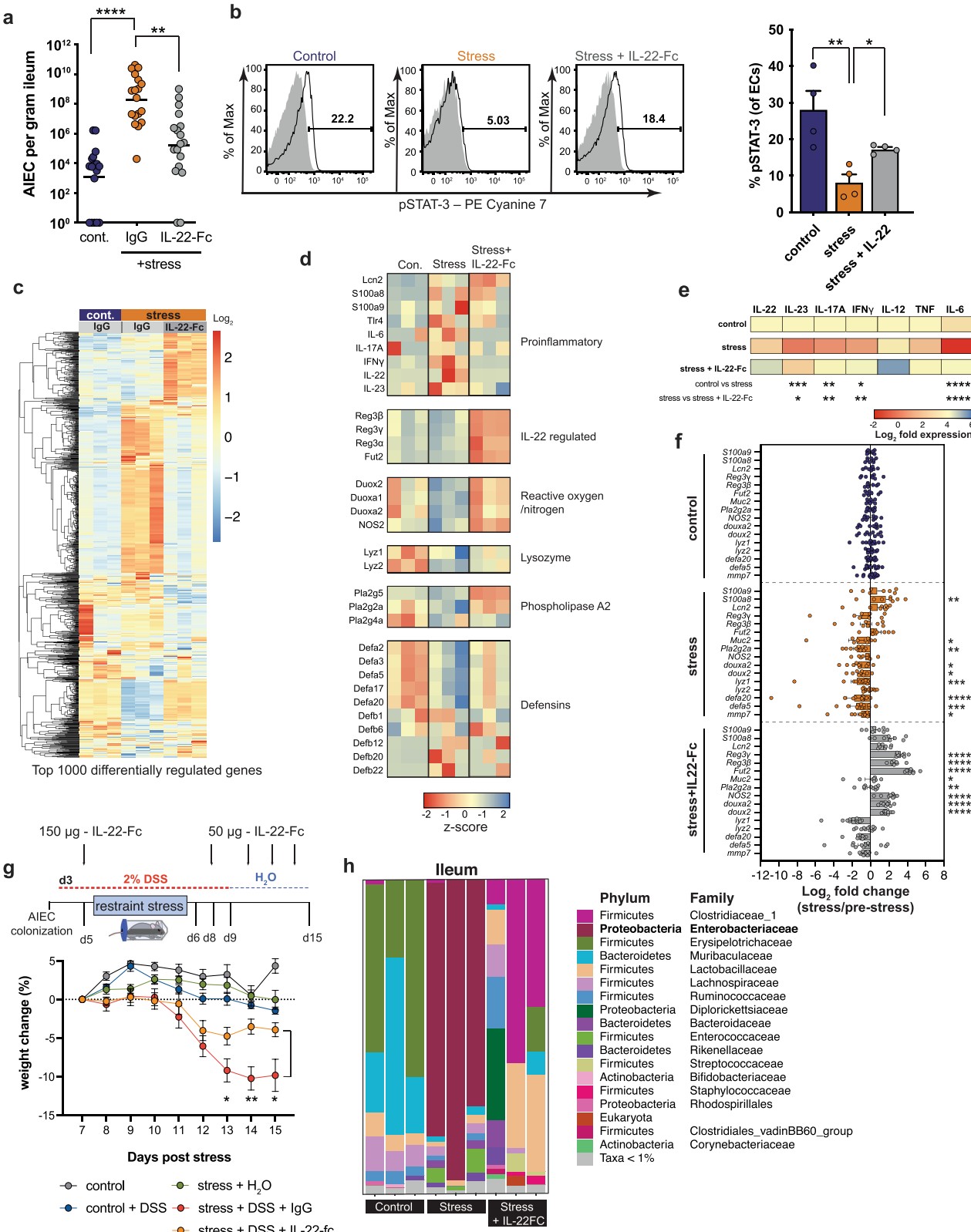

enriched in CD patients[7-9]. For instance, an enrichment of *Enterococcus faecalis* is strongly associated with disease severity in CD patients[74], and this species was also enriched following exposure to psychological stress. *Bifidobacterium*, which are commonly associated with intestinal health and generally have decreased abundance in CD, were also decreased following overnight stress.

Psychological stress has traditionally been considered anti-inflammatory in nature, owing to the production of glucocorticoids[50,75,76]. Our findings, in line with a recent publication[43], demonstrate that stress induces a mixed inflammatory response. In our model of psychological stress, the pro-inflammatory features appear to play a dominant role in shaping the microenvironment that favors the expansion of AIEC. For

**Fig. 7 IL-22 delivery corrects stress-induced intestinal dysbiosis. a** AIEC tissue burdens collected from the ileum of starved ($n = 18$), IgG treated stress ($n = 18$), and IL-22 treated stress ($n = 18$) mice. Significance was determined by one-way ANOVA, not corrected for multiple comparisons, control:stress, $p < 0.0001$; stress:stress + IL-22, $p = 0.0030$. **b** Representative FACS plots of phospho-STAT-3$^+$ cells from isolated epithelial cells. Percentage of phospho-STAT-3 on ileal epithelial cells of starved ($n = 4$), IgG treated stress ($n = 4$), and IL-22 treated stress ($n = 4$) mice as determined by flow cytometry. Significance was determined by one-way ANOVA, not corrected for multiple comparisons, control:stress, $p = 0.0017$; stress:stress + IL-22, $p = 0.0392$. **c** RNA sequencing analysis of the top 1000 differentially regulated genes from ileal samples of AIEC-colonized starved ($n = 3$), stressed ($n = 3$), and IL-22 treated, stressed ($n = 3$) mice (GSE180342). **d** Z-score values from RNA sequencing analysis of innate immune pathways from ileal samples of AIEC-colonized starved ($n = 3$), stress ($n = 3$), and IL-22 treated stress ($n = 3$) mice. **e** RT-qPCR analysis of cytokine expression from ileal samples of AIEC-colonized starved ($n = 8$), IgG treated stress ($n = 8$), and IL-22 treated stress ($n = 8$) mice. Control and stress samples are the same as Fig. 3b. Significance was determined by two-way ANOVA, not corrected for multiple comparisons. IL-23, control:stress, $p = 0.004$, stress:stress + IL-22-Fc = 0.0185; IL-17A, control:stress, $p = 0.0027$, stress:stress + IL-22-Fc = 0.0044; IFN-γ, control:stress, $p = 0.0131$, stress:stress + IL-22-Fc = 0.0049; IL-6, control:stress, $p < 0.0001$, stress:stress + IL-22-Fc < 0.0001; **f** RT-qPCR analysis of AMP expression from ileal samples of AIEC-colonized starved ($n = 8$), IgG treated stress ($n = 8$), and IL-22 treated stress ($n = 8$) mice. Significance was determined by two-way ANOVA, not corrected for multiple comparisons. Stress significance is relative to control. Stress + IL-22-Fc significance is relative to stress. Control and stress samples are the same as Fig. 4b. Control:stress (S100a8, $p = 0.0055$; Muc2, $p = 0.0113$; Pla2g2a, $p = 0.0053$; douxa2, $p = 0.0146$; doux2, $p = 0.0394$; Lyz1, $p = 0.0010$; defa20, $p < 0.0001$; defa5, $p = 0.0005$) Stress:stress + IL-22-Fc (Reg3g, $p < 0.0001$; Reg3b, $p < 0.0001$; Fut2, $p < 0.0001$; Muc2, $p = 0.0199$; Pla2g2a, $p = 0.0079$; Nos2, $p < 0.0001$; Douxa2, $p < 0.0001$; Doux2, $p < 0.0001$). **g** Schematic representation of DSS treatment schedule. The graph depicts weight change normalized 2 days post stress (control $n = 4$, control + DSS $n = 7$, stress + H$_2$O $n = 4$, stress + DSS $n = 7$, stress + DSS + IL-22-Fc $n = 6$). Significance between stress + DSS and stress + DSS + IL-22-Fc was determined by multiple unpaired two-tailed $t$-tests, not corrected for multiple comparison. Day 7, $p = 0.042223$; day 8, $p = 0.004173$; day 9, $p = 0.032142$. **h** Taxonomy plots of 16s rRNA sequencing of the ileal contents of starved ($n = 3$), IgG treated stress ($n = 3$), and IL-22 treated stress ($n = 3$) mice. (*$p \le 0.05$; **$p \le 0.01$; ***$p \le 0.001$; ****$p \le 0.0001$). Error bars represent SEM and the line in CFU graphs indicates the geometric mean of the group.

example, mice exposed to stress had reduced expression of genes associated with barrier function, including tight junctions and mucins. The encroachment of AIEC towards the epithelium and dissemination to systemic sites that we observed following stress is likely facilitated by these weakened physical barriers, which further provokes the host inflammatory response. Indeed, we observed more pronounced barrier defects, increased pro-inflammatory cytokine expression, and dysbiosis in AIEC-colonized mice exposed to psychological stress compared to AIEC-naive mice exposed to the same stressor. Thus, the combination of inflammation-tolerant bacteria like AIEC, and the mucosal defects brought about by stress, appear to create a tipping point in gut homeostatic balance that favors inflammation.

Interestingly, while nutritional immunity was necessary, it was not sufficient to phenocopy the ileal dysbiosis induced by stress. Indeed, psychological stress independently impaired IL-22-driven protective mucosal immunity against AIEC. A pivotal role for IL-22 was revealed using IL-22 depletion and administration of IL-22-Fc to probe the functional consequences of this pathway during stress. Administration of IL-22-Fc functionally restored the expression of genes downstream of IL-22 signaling including *fut2* and antimicrobial proteins which appeared to have a functional effect in preventing stress-induced AIEC expansion and in correcting the ileal dysbiosis that occurred following stress. Interestingly, recent clinical studies are evaluating the use of IL-22 in the treatment of IBD patients (Clinical Trial: NCT03558152, NCT02749630). When ileal explants taken from mice exposed to stress were stimulated with IL-23, these tissues failed to produce IL-22 whereas ileal tissues from unstressed mice showed a robust IL-22 signature, suggesting that the IL-22 producing cell population was depleted following psychological stress. Indeed, the gut of mice exposed to psychological stress had a significant reduction in the number of CD45$^+$CD90$^+$ cells in the small intestine following psychological stress, which we directly linked to stress because this cell population was restored by blocking glucocorticoid signaling.

Although psychological stress impaired IL-22-dependent host protection, nutritional immunity appeared to remain intact, and in fact was required for AIEC expansion. The host protein calprotectin sequesters zinc, manganese, calcium, and iron under states of inflammation[59,77]. In response to host nutritional

immune pressure, some Enterobacteriaceae have evolved mechanisms to exploit nutrient limitation to outcompete commensal microbes[27,28,78]. For example, *Salmonella* expresses a high-affinity zinc transporter, ZnuABC, and can thrive in the presence of calprotectin-mediated nutritional immunity[77]. Interestingly, AIEC appears to also thrive in inflammatory environments[13,14] and has acquired the ZnuABC transporter, likely providing a fitness advantage in states of nutritional immunity. Given the essentiality of iron for bacterial replication, the host has multiple mechanisms to limit the availability of iron in its various forms. Iron is typically transported complexed with heme and is a common target of bacterial species[79,80]. As such, the host sequesters free-heme in pathophysiological settings[22,29]. Accordingly, we found that both HPX and haptoglobin were upregulated during psychological stress exposure, potentially limiting iron availability to bacterial species[22,29,81]. Enhanced recruitment of neutrophil-like CD11b$^+$Gr1$^+$ cells in our model was accompanied by robust induction of neutrophil-derived Lcn2 following stress. The host releases Lcn2 to limit the acquisition of iron-bound enterobactin, preventing its reuptake by commensal strains of *E. coli*[23]. Unlike most commensal strains, many AIEC encode the biosynthetic and secretion machinery for salmochelin, a glycosylated variant of enterobactin that evades Lcn2 sequestration[27,60,82] and provides a competitive advantage to *Salmonella*[27,82]. Indeed, AIEC derived a salmochelin-dependent competitive advantage in stressed mice, but not in unstressed mice, indicating that psychological stress creates a competitive gut environment that appears to benefit microbes that have evolved mechanisms to evade nutritional immunity. Interestingly, we found that neither LPS-mediated induction of iron limitation, nor CD90 depletion, in isolation, was sufficient to phenocopy the AIEC expansion seen during stress. Instead, our data are consistent with a combinatorial effect of psychological stress whereby the combined activation of nutritional immunity and immune cell attrition support the intestinal dysbiosis seen following psychological stress.

Overall, our study shows that psychological stress creates a beneficial environment for AIEC, a CD-associated pathobiont in the gut. Given that the pathological changes observed following psychological stress are augmented in the presence of AIEC, this work establishes a rationale for future studies to dissect the

relative contributions of the microbiome and psychological stress on the gut environment. In our current study, AIEC appears to derive this benefit by evading host nutritional immunity while taking advantage of an impaired induction of IL-22-mediated host defenses that rely on antimicrobial proteins and barrier maintenance. Thus, in the presence of pathobionts that efficiently evade nutritional immunity, this aberrant stress-induced host response provides a promiscuous niche for their unregulated expansion. The ability of IL-22 treatment to correct both mucosal host defenses and prevent *E. coli*-dominated ileal dysbiosis provides compelling rationale for continued investigation of this intervention. This work reveals insight into the role that psychological stress plays in disease expression of CD. Uncovering the interactions between microbes, the host immune system, and epithelial host defenses will lay the biological underpinnings that guide preventions and therapies that address unmet clinical needs for CD management.

## Methods

All research was reviewed and approved by the Animal Review Ethics Board (AUP# 20-12-41) at McMaster University and conducted in accordance with standards set by the Canadian Council of Animal Care.

**Animal infections and treatments**. Six- to eight-week-old male C57BL/6N mice were purchased from Charles River Laboratories (#027, C57BL/6N, QC, CAN). TNFKO were originally purchased from Jackson Laboratory (#003008, ME, USA) and were bred inhouse. All mice were housed in Level 2 biohazard containment under specific pathogen-free barrier conditions and maintained on a 12 h light:12 h dark cycle, which was temperature-controlled (21 °C), 30–50% humidity. One day prior to colonization with AIEC, mice were treated with 20 mg of streptomycin (Sigma-Aldrich, ON, CAN) by oral gavage. Mice were infected with $2 \times 10^9$ colony forming units (CFU) of AIEC strain NRG857c[83] in 0.1 ml sterile phosphate buffer saline (PBS). NRG857c was routinely grown shaking in Luria broth (LB, Sigma-Aldrich) with chloramphenicol (34 µg/mL) and ampicillin (200 µg/mL). NRG857c $\Delta iroB$ was grown as described above, with the addition of gentamicin (20 µg/mL). RU486 (Sigma-Aldrich) was dissolved in DMSO and delivered one hour prior to stress at a dose of 50 mg/kg, control mice were given an equivalent volume of DMSO. Anti-CD90 treated mice were treated every other day intraperitoneally (i.p.) with 200 µg of anti-CD90 (clone 53 2.1, BioXcell), starting on the day of infection. The IL-10R blocking antibody (clone 1B1.3A, BioXcell, NH, USA) was administered at 200 µg/mouse in 200 µL PBS and delivered i.p. 1 h prior to stress. In some cases, LPS (Sigma-Aldrich Lipopolysaccharides from *Salmonella enterica*) was administered i.p. at 0.25 mg/kg. IL-22 was blocked using 150 µg anti-IL-22 antibody (8E11; Genentech) in PBS delivered i.p. daily. IL-22-Fc (PRO312045; Genentech) was delivered to mice at a dose of 150 µg in PBS delivered i.p. A matched IgG control was given to control mice. For long-term anti-IL-22 treatment a dose of 150 µg/mouse was administered i.p. every other day for 11 days. All experiments were composed of 4 mice per group and the total number of mice used are represented in the figure legends. Male mice were used to minimize the confounding effects of the female estrus cycle on RU486 treatment. All preliminary results were confirmed in female mice.

**Restraint stress**. Stress-exposed mice were placed in well-ventilated 50 mL conical tubes (20 holes per tube) for 16 h during their dark cycle. Matched control mice were deprived of food and water for 16 h[44,45]. For tissue CFU enumeration, mice were immediately sacrificed, and samples were collected in sterile PBS. In some experiments, mice were allowed to recover, and fecal samples were collected at 6 and 24 h following restraint stress. In repeated stress experiments, mice were subjected to weekly overnight restraint stress and allowed to recover, with fecal samples collected prior to and at 6 and 24 h post-stress. In one experimental series, mice were sacrificed after 4 h of stress and samples taken for RT-qPCR analysis.

**Bacterial enumeration**. Fecal or tissue samples were collected in 1 mL sterile PBS. The small intestine was divided into four equal segments, as previously described[84]. Segment 1 denotes the segment closest to the stomach and segment four closest to the cecum. Segment 4 represents the ileal region of the small intestine and is used interchangeably throughout. Whole tissue from four segments from the small intestine, as well as the cecum and colon, were collected at the time of sacrifice and placed in 1 mL sterile PBS and homogenized using a Mixer Mill. Homogenized samples were serially diluted and plated on LB agar supplemented with ampicillin (200 µg/mL) and chloramphenicol (34 µg/mL). Liver samples were processed similarly and plated undiluted on un-supplemented LB agar for the enumeration of total viable bacteria.

**mRNA assessment by reverse-transcriptase quantitative PCR**. Tissue samples were taken from the distal 1 cm of ileum (Segment 4) and placed in TRIzol (Invitrogen, ON, CAN). Total RNA was extracted and converted to complementary DNA (cDNA) using a one component cDNA SuperMix (Quanta Biosciences, MA, USA). Quantitative PCR was performed on a Lightcycler 480 running software v1.5.1.62 SP2 (Roche, QC, CAN), using SYBR green SuperMix (Quanta Biosciences). All primers used in the study are listed in Supplementary Table 1. The cycling conditions were 95 °C for 5 min and 50 cycles of 95 °C for 10 s, 55 °C for 30 s, and 72 °C for 20 s. Gene expression was evaluated using the $2^{-\Delta\Delta CT}$ method and samples were normalized to the housekeeping gene, *RPLP0*, and expressed relative to the control starved mice.

**Quantification of biomolecules**. Liver LPS levels were enumerated using a Pierce™ Chromogenic Endotoxin Quant Kit (Thermo Fisher Scientific, MD, USA). Values are represented as endotoxin units (EU). Mouse Lcn2/NGAL and S100a8 in tissue homogenates were enumerated using a DuoSet ELISA (R&D Systems, MN, USA) as per the manufacturer's protocol. Numerical values were determined using an 8-point standard curve of known concentrations. Mouse IL-22 in ileal explant supernatant was enumerated using an ELISA (ThermoFisher) as per the manufacturer's protocol. Numerical values were calculated using an 8-point standard curve of known concentrations. Mouse corticosterone in the serum was enumerated using a Corticosterone Parameter Assay Kit (R&D Systems) as per the manufacturer's protocol. Numerical values were determined using a 6-point standard curve of known concentrations, using an EnVision 2104 Multilabel Reader (v1.13). Serum samples were collected after 16 h of stress or overnight starvation and analyzed at Eve Technologies (Alberta, Canada) for cytokine and chemokine determinations using multiplex analyses. Samples are represented using a mean z-score.

**16S rRNA sequencing**. Luminal contents were scraped from the various intestinal segments taken from control mice (ad libitum food/water), control (food/water restricted), stressed, and stressed with IL-22-Fc-treated mice and placed in a sterile collection tube containing NaPO₄ and guanidine thiocyanate/EDTA. All mice arrived at the animal facility in the same shipping crate, were caged in the same bedding, and treated under the same conditions prior to stress. Samples were processed by the Farncombe Metagenomics Facility (McMaster University). 16s rRNA v3v4 regions were sequenced using an Illumina sequencer. Taxonomy was assigned to the output sequences using the Silva reference dataset (v 132) using the R package DADA2 (v1.16). Taxonomical plots were created using the R packages phyloseq (v1.32.0) and phytools (v0.7-20) and plotted using ggplot2 (v3.3.0). Relative proportional change was calculated based on the frequency of bacterial species present in the starved control mice compared to overnight stressed mice. Values were plotted as $\log_{10}$ change in abundance. Community diversity was assessed by calculating the Shannon diversity index using the R package, vegan (v2.5-6). The factoextra package (v1.0.7) was used to calculate the eigenvectors for principal component analysis and plotted using ggplot2.

**LDA effect size (LefSE)**. LefSE was calculated using $p < 0.05$ for the factorial Kruskal–Wallis test and $p < 0.05$ for the pairwise Wilcoxon test. A *LefSE* score > 2 was used as the threshold cutoff. All values were determined and plotted using the Galaxy server https://huttenhower.sph.harvard.edu/galaxy/[85].

**RNA sequencing of bulk intestinal tissue**. Tissue was isolated from the distal 1 cm of ileum and placed in TRIzol. RNA was extracted and frozen in DEPC water prior to shipment to Genewiz for RNA sequencing. Ribosomal RNA was removed using a Ribo-Zero Gold Kit (Illumina, San Diego, CA, USA), and the sequencing library prepared using a NEBNext Ultra RNA Library Prep Kit from Illumina. Sequencing analysis was done using an Illumina HiSeq using a $2 \times 150$ bp paired-end configuration, single index, per lane. One mismatch was allowed for index sequence identification. Transcripts were quantified using Salmon (v1.3.0)[86] and differential expression levels were evaluated using the R package DeSeq2 (v1.28.1)[87]. Heatmaps were generated using the R package pHeatmap (v1.0.12)[88]. Raw RNA sequencing data has been archived in NCBI under accession GSE180342.

**Intestinal permeability measurements**. Ussing chambers were used to evaluate the permeability of ileum sections ex vivo following overnight starvation or stress. Paracellular permeability was assessed using 6 µCi/mL of $^{51}$Cr-EDTA (type II; Perkin Elmer) probes. $^{51}$Cr-EDTA was measured in a liquid scintillation counter and reported as percent recovery/cm²/h.

**Dextran sodium sulfate (DSS)-induced colitis**. Mice were started on 2% DSS in drinking water ad libitum (40,000–50,000 MW, Thermo Scientific) on day 3 post-infection with NRG857c and mice were stressed for 16 h on day 5. Mice were switched back to normal drinking water on day 8 or 9. Body weight was recorded daily. In cases where mice were treated with IL-22-Fc (150 µg prior to stress, 50 µg every 2 days following stress), matched groups received an equivalent amount of matched IgG (Genentech).

**Construction of mutants**. Lambda Red recombination[89] was used to delete the *iroB* gene with modified plasmids that allowed for selection with gentamicin. Primer sequences used for mutant generation and screening are listed in Supplementary Table 2. In brief, primer pairs containing extensions complementary to 100 bp in-frame regions at the 5′ and 3′ ends of each gene of interest were used to PCR amplify the gentamicin resistance cassette from pCDF_GmFrt and generate knockout constructs. PCR products were purified and transformed into NRG857c harboring pKD46_km. Transformants were selected on LB-agar containing 20 μg/mL gentamicin and successful deletion of genes was confirmed through PCR and sequencing (Genewiz, South Plainfield, NJ).

**Competitive infections of mice**. Wild type or mutant AIEC NRG857c were grown overnight in LB with selection as described above. For competitive infections, mice were treated with streptomycin (20 mg) 1 day before infection and infected with $2 \times 10^9$ CFU of an equal mixture of competing strains. Mice were exposed to overnight restraint stress or food/water deprivation on day 2 following infection. Bacterial CFU was enumerated in tissue samples as described above. To account for in vivo bottlenecks, the competitive index was normalized to the average ratio of wild type to mutant AIEC within fecal samples taken at the time of stress.

**Cell isolation and flow cytometry**. Epithelial cell and lamina propria cell isolation —Ileum was extracted from control, stressed, and IL-22-treated stressed mice at the time of sacrifice and placed in cold PBS. Feces was removed and samples were opened longitudinally before being cut into 1 cm segments and placed in 15 mL tubes containing 10 mL PBS. Samples were washed in PBS, placed in 10 mL of 5 mM EDTA and 2 mM DTT, and incubated at 37 °C. Samples were rotated at 15 rpm in a hybridization oven for 30 min and filtered through a 100 μM filter. Following epithelial cell isolation, cells were further digested for the isolation of lamina propria cells in 12.5 μg/mL Liberase TM (Sigma-Aldrich) and 1 mg/mL DNase I (Sigma-Aldrich) at 37 °C. Control samples are comprised of 2 pooled mice and all stress samples are comprised of four pooled mice. Samples were rotated at 8 rpm in a hybridization oven for 10 min and filtered through a 100 μM filter. Samples were enriched for mononuclear cells using a 40% Percoll gradient. Isolated cells were incubated with CD16/32 (Fc Block) for 10 min prior to staining. For epithelial cell analysis, cells were stained for 30 min with CD45—eflour450 (1:400), EpCam—APC (1:400) in 2% FBS/PBS at 4 °C. Following surface staining, cells were washed, fixed, and permeabilized using a *Foxp3* transcription factor staining buffer (Thermo Fisher) for 20 min at room temperature. Following fixation, cells were stained for 30 min with 5 μL of pSTAT-3-Pe Cyanine 7 (Thermo Fisher), washed, and resuspended in 2% FBS/PBS. Isolated lamina propria cells were stained for 30 m with a cocktail of lineage exclusion markers (Ter119, Gr1, CD3, CD11b, B220)—FITC (1:400), CD90—APC eflour 780 (1:600), CD45—eflour 450 (1:600), TCRβ—SuperBright785 (1:600), CD4—APC eflour 780 (1:400), CD8—Alexa Fluor 532 (1:400) for lymphocyte analysis, or CD45—PercpCy5.5 (1:600), CD11b—eflour 605 (1:800), Gr1—FITC (1:800) for neutrophil-like cells. Cells were stained intracellularly for roryt—APC (1:50), and GR—PE Cyanine 7 (1:20) for 30 min in 1 × Perm/fix buffer at room temperature. For Annexin V staining isolated lamina propria cells were washed with PBS and Annexin binding buffer (ThermoFisher) and stained for 25 min at 37 °C with CD45—PercpCy5.5 (1:600), CD90—PE Cyanine 7 (1:600), Annexin V—APC (5 μL/test). Cells were washed and samples were evaluated using a LSRII flow cytometer (BD Biosciences) using BD FACS Diva (v9.0) and analysis was performed using FlowJo (v9, BD Biosciences). Gating strategies are found in Supplementary Fig. 9.

**Explants**. Ileum was extracted from mice at the time of sacrifice and the fecal content was cleared and washed with cold PBS. After washing, 1-cm sections of ileum were treated with 100 μg/mL of gentamicin for 1 h, washed and cultured for 24 h in complete RPMI (10% FBS, 0.1 mM MEM nonessential amino acids, 1 mM sodium pyruvate, 100 U/mL penicillin, 100 μg/mL streptomycin, and 10 mM HEPES) with or without rIL-23 (20 ng/mL) and supernatants were collected after 24 h.

**Immunohistochemistry**. Samples of mouse ileum were fixed in 10% formalin following stress, paraffin embedded, and stained for GR1 at the Histology Services (McMaster University). IHC quantification represents an average of four sections per sample by an independent reviewer blinded to the experimental treatment groups. All slides were examined on a Zeiss Axio Imager with ×10 objective.

**Statistical analysis**. A two-tailed Mann–Whitney test was applied to experimental data where only two groups were present. A standard one or two-way ANOVA was used for comparisons of three or more groups depending on the experimental design, no correction for multiple comparisons was performed unless otherwise stated. Correlation analysis was done using a two-tailed Spearman correlation analysis. Values were considered statistically significant for $p < 0.05$. Statistical analysis was performed using Prism GraphPad (v8). Raw data was collected in sheets using Excel (v16.53).

**Reporting summary**. Further information on research design is available in the Nature Research Reporting Summary linked to this article.

## Data availability

The RNA sequencing data have been deposited in NCBI Gene Expression Omnibus under the accession code GSE180342. All additional data available within the manuscript, its figures and supplementary files, has been included as individual datasheets in the file Source Data.zip. Silva reference dataset (v 132), available at https://www.arb-silva.de/no_cache/download/archive/release_132/. Source data are provided with this paper.

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

## Acknowledgements

This work was supported by grants from the Canadian Institutes of Health Research (CIHR; PJ4-175368 and MOP-136968; B.K.C.); Crohn's and Colitis Canada (B.K.C., E.V.); CIHR Postdoctoral Fellowship Awards (C.R.S., W.E.); the CIHR Canada Graduate Scholarship (A.A.P.); and the Canada Research Chairs program (B.K.C., E.V.).

## Author contributions

Conceptualization: C.R.S. and B.K.C. Methodology: C.R.S., A.A.P., and B.K.C. Investigation, validation, formal analysis: C.R.S. and A.A.P. Visualization and writing—original draft: C.R.S., A.A.P., and B.K.C. Investigation: J.D., W.E., J.J., and E.F.V. Project administration, supervision, and funding acquisition: B.K.C. Writing—review & editing: B.K.C. All authors have read and approved the final version of the manuscript.

## Competing interests

The authors declare no competing interests.
