## [Peer Review File · Nature Communications]

Psychological stress impairs IL22-driven protective gut mucosal immunity against colonising pathobiontsREVIEWER COMMENTS

Reviewer #1 (Remarks to the Author):

The study „Psychological stress impairs IL-22-driven protective mucosal immunity against CD-associated pathobionts“ by Shaler and Parco et al. investigates the impact of psychological stress on intestinal host-microbiota homeostasis with a focus on host control of AIEC, a known CD-associated pathobiont. The authors show that stress induces a dysbiotic shift in the microbiota, which is dominated by an outgrowth of Enterobacteriaceae. Building on this, the authors specifically study AIEC colonization under stress, revealing that stress impairs host control of AIEC, by depleting protective IL-22-producing immune cell populations in the intestine.

In general, the manuscript is well written and the figures are nicely structured and illustrated, making it easy to read and understand. Furthermore, the work provides important and interesting new insights into the role of psychological stress on host-microbiota interactions. Given the fact that disturbances in host-microbiota crosstalk are key driver of IBD, and psychological stress is known to be associated with increased IBD activity, the manuscript addresses important aspects of disease pathology by studying the links between stress and host-microbiota dysbiosis. Therefore the study is likely to be of considerable interest for the readers in the field.

In contrast, my major concern is that the strong focus on AIEC makes it difficult to precisely dissect the effects of stress itself, the stress-induced dysbiotic microbiota and the colonization with AIEC on the host, which weakens the overall quality of the paper in its current state.

Specific points:

- The authors report that stress induces a dramatic microbiota dysbiosis, which is accompanied by defects in barrier functions and increased inflammation. However, the consequences as well as the underlying mechanisms of this direct stress-induced disruption of host-microbiota homeostasis are not comprehensively studied throughout the manuscript. Instead, the authors focus very early on the stress + AIEC colonization model, which introduces another layer of complexity, namely the individual effect of AIEC on the host within an already dysbiotic microenvironment. Therefore, I suggest to separate both models (naïve vs +AIEC) more clearly, which could help to delineate the individual impact of stress and AIEC much better. By this, the authors could first describe the role of stress in creating an (inflammatory) microenvironment that predisposes for pathobiont colonization and outgrowth before showing how AIEC exploits this susceptible state. On the same lines, important questions remain to be answered: What is the individual impact of the stress-induced dysbiotic microbiota on the host? Also, is the stress-induced dysbiotic microbiota needed for AIEC pathogenicity? To answer these questions experiments with germ-free mice could help to define the specific role of the microbiota for the stress-response.
- Figure 1: What are the dynamics of the microbial dysbiosis and outgrowth of Enterobacteriaceae? Does the dysbiotic microbiota revert back to homeostasis like the AIEC burden shown in Fig. 2b/c?
- Figure 2c: Do repetitive stress and AIEC outgrowth result in some sort of spontaneous intestinal inflammation?
- Figure 2e: I am wondering why the 16S data from AIEC-colonized naïve and starved animals don't show any abundance of Enterobacteriaceae (except for one mouse per group)? Should the presence of AIEC not also be reflected in the sequencing data?
- Figure 3c: What is the impact of stress alone (naïve, w/o AIEC colonization) on the barrier genes?
- Figure 3e-h: What is the impact of stress alone (naïve, w/o AIEC colonization)?
- Figure 3i: Is TLR4 inhibition also sufficient to contain Enterobacteriaceae outgrowth (naïve, w/o AIEC colonization)?

- Figure 3j: Again, if this is data from the naïve mice, is this also true for the AIEC colonization model?

Moreover, the FACS data don't look very convincing. 80% of all CD45+ ileal lamina propria cells are CD11b+Gr1+ neutrophils?? This appears unrealistic to me. There is also a very high variance within the stress group ranging from 10% to 80%. Maybe the authors could repeat these measurements and include more markers and stain for more cell types, such as T cells, B cells etc. to get a more comprehensive picture.

- Figure 4: How is the stress response affecting nutritional immunity in naïve mice (w/o AIEC colonization)? In this context, how can the outgrowth of endogenous Enterobacteriaceae be explained?

- Figure 5b: What are the total numbers? The authors should also include more inclusion markers for T cells and ILCs. For example: CD90 plus lineage negativity is only a very crude definition of ILCs. Additionally, the authors could use CD127 for ILCs and lineage-defining transcription factors to even define the major T and ILC subsets (Th1, Th2, Th17 and ILC1, ILC2, ILC3).
In general: How is stress mediating this depletion of lymphocytes? Is it inducing apoptosis in lymphocytes? This should be experimentally addressed (e.g. by staining for apoptosis markers) and discussed.

- Figure 5c/d: Is the cytokine measurement done directly ex vivo or after some sort of restimulation? Usually, to detect cytokine expression, restimulation with PMA/Iono or cytokines, e.g. IL-23/IL-1 β for IL-22, is needed. What is exactly depicted in 3c, three individual mice? If yes, why is the background and gating always different? Representative dot plots for IL-17 and IL-22 are missing. The authors should also discriminate between T cells and ILCs here.

- Figure 5d: The authors show a decrease in IFN γ + and IL-17+ producing CD90+ immune cells upon stress. However, in Figure 3b they show an increased expression of IFN γ , IL-17 etc. in the tissue. Given that T cells and ILCs are the major effector immune cells in the gut, how can these contradictory findings be reconciled?

- Figure 5e-h: What is the major source of IL-22 in this model? Are Th17 cells or ILC3s or both mediating protection from AIEC colonization?

- Figure 6c: It would be very interesting to compare the global ileal transcriptome of stressed AIEC-colonized mice also to the one of stressed naïve mice. Thereby, the effect of stress vs. AIEC colonization could be dissected and AIEC-dependent and AIEC-independent stress-induced pathways could be revealed.

Reviewer #2 (Remarks to the Author):

Psychological stress is known to influence the disease in Crohn's disease (CD) patients. It has been reported that psychological stress can be a driver of gut dysbiosis. However, the effect of psychological stress on CD-associated dysbiosis and its relation to the pathogenesis of disease remain poorly understood. In this manuscript, Shaler et al. investigated the impact of psychological stress on the expansion of CD-associated adherent-invasive E. coli (AIEC). The authors demonstrated that restraint stress resulted in the gut dysbiosis accompanied by the expansion of Enterobacteriaceae in the gastrointestinal tract (particularly in the ileum) in SPF mice. Next, the authors used the AIEC colonization model and confirmed that psychological stress promoted persistent colonization by AIEC. The enhanced colonization by AIEC exacerbated DSS-induced colitis in stressed mice. Also, the increased colonization by AIEC impaired the gut barrier integrity and mucosal inflammatory responses. Psychological stress-induced nutritional immunity, which conferred the competitive outgrowth of AIEC over competing commensals. The authors then focused on IL-22 signaling as a mechanism by which AIEC expanded in stressed mice. Psychological stress-induced glucocorticoids reduced the number of IL-22-producing lymphocytes (T cells, ILCs), subsequently causing the outgrowth of AIEC. Consistent with this notion, reconstitution of IL-22 signaling restored the gut microbiota and protected mice from colitis.

Linking AIEC expansion and psychological stress is an intriguing concept and potentially crucial for understanding the pathogenesis of CD. The experiments are extensive. However, the merit of the study is significantly dampened by several shortcomings, particularly in the mechanistic depth.

Specific comments:

1. AIEC infection model is useful to study the pathogenic role of this particular bacterium. However, the model is somewhat artificial. The authors nicely showed that psychological stress leads to gut dysbiosis with the massive expansion of Enterobacteriaceae (likely dominated by *E. coli*) in Fig. 1. Does the expansion of indigenous Enterobacteriaceae influence the susceptibility to colitis? Were indigenous AIEC-like bacteria (e.g., salmochelin-expressing) are dominated within the Enterobacteriaceae?
2. There was a discrepancy in the regulation of IL-22 signaling. In Fig. 3b, IL-22 and other cytokines (e.g., Th1, Th17-related cytokines) were significantly up-regulated in AIEC/stress mice. However, in Fig. 5, the number of IL-22 (and IL-17A, IFN γ)-producing lymphocytes was reduced in the stressed group. Since the authors claimed that reducing IL-22-producing cells is a crucial mechanism that drives AIEC expansion, this is a critical flaw.
3. The authors claimed that impaired IL-22 signaling by psychological stress is the critical mechanism of gut dysbiosis (AIEC domination). However, upstream and downstream mechanisms of IL-22 signaling defect were insufficiently provided. The authors claimed that glucocorticoid signaling leads to the loss of IL-22-producing cells. Although RU486 treatment restored the number of those cells, the precise mechanism remains unclear. Also, how does IL-22 shapes healthy microbiota was unclear. Since IL-22 is considered as the central pathway that connects psychological stress and gut dysbiosis, more detailed mechanistic validation is required to support the authors' claim.
4. Although the microbial composition of control and stress mice was nicely analyzed, baseline microbial composition was not shown. It is essential to show the baseline microbiota is standardized in all groups; otherwise, the cage effect cannot be ruled out. At least, a detailed description of how gut microbiota composition was normalized among groups needs to be provided in the method section. Were littermates used? All mice are co-housed or performed mixed-bedding prior to the experiments and then randomized?
5. The staining pattern of neutrophils is a little odd. Almost all CD45+ cells in the ileal LP cells are neutrophils in the stress group. The authors should confirm this phenotype by histology (HE staining, immunostaining for neutrophils).
6. Based on the experimental procedures provided in Figures, only male mice were used in all animal experiments. Justification needs to be provided on why the authors used only one sex.
7. Fig. S6 was related to neither psychological stress nor AIEC colonization. Thus, this data set is not suitable for this study.

Reviewer #3 (Remarks to the Author):

This manuscript reports that acute psychological stress results in a bloom of ileal Enterobacteriaceae and, moreover, makes mice highly prone to colonization by adherent-invasive *E. coli*, AIEC, which are known to associate with, and thought to promote severity of, Crohn's disease. Accordingly, stress and AIEC synergize to exacerbate DSS-induced colitis. Mechanistically, the phenotype appears to result from altered leukocyte trafficking that impairs IL-22 production by resident lymphocytes.

Wow! This is some of the most striking changes in microbiota that I have ever seen, resulting from a psychological event, albeit a very extreme one. I think the implications of this are quite broad and go well beyond the phenotypic readout used here. The mechanism remains far from

understood but, nonetheless, a very solid start has been provided. I think the striking changes in microbiota warrant publication quickly but the mechanistic conclusions could use some better characterization or should at least be toned down. Specific comments are offered.

1. Re sampling/analysis of microbiota from various intestine segments, please specify precisely how sample was collected. Is it luminal content that was assayed? Luminal content excluded and only adherent bacteria assayed? Both? I don't really have a concern with what is the specific answer to this query but it needs to be precisely specific so other can reproduce the work.
2. I don't understand how the data is displayed in Fig 1 E-F. In particular, what is the X-axis? Why is bifido on left and fecalbacterium on right. I see empty tick marks. Should there be numbers below them?
3. The ultimate phenotypic readout for impacts of stress and consequently AIEC colonization is exacerbated DSS colitis. This seems slightly incongruous in that stress predominantly impacts the ileum whereas DSS is usually purported to largely affect colon. Hence, it would be helpful to know if stress/AIEC made the SI prone to DSS-induced disease or largely impacted colon disease, or just acted in a general way on a systemically-influenceable parameter like body weight.
4. It is clear that overnight withholding of food/water is not sufficient to increase Enterobacteriaceae and proneness to AIEC but is it necessary? Could a mode of stress without starvation suffice? This does not necessarily need to be addressed experimentally at this time but it should be discussed.
5. The notion that increased TLR4 signaling drives AIEC is interesting as, intuitively, one might have more reasonably presumed that increased TLR4 signaling resulted from AIEC, but certainly both could be true. I suggest this point be discussed. But, in any case, the use of the inhibitor lacks a proper control. TLR4 KO mice are readily-available (on a C57 background or C3HeJ) and this the need for TLR4 in driving stress-induced should be investigated via such mice. These mice can also be used to check the specificity of TAK-242, which is far assured.
6. The impairment of IL-22 expression resulting from stress is quite impressive. To what extent is acute loss of IL-22 induction sufficient for AIEC colonization and resulting phenotype? While importing IL-22-KO mice might take awhile, I note their supplier of Fc-IL-22 (Genentech) routinely provides large amounts of neutralizing anti-IL-22 Mab.

Response to Reviewer's Comments

Reviewer #1 (Remarks to the Author):

The study "Psychological stress impairs IL-22-driven protective mucosal immunity against CD-associated pathobionts" by Shaler and Parco et al. investigates the impact of psychological stress on intestinal host-microbiota homeostasis with a focus on host control of AIEC, a known CD-associated pathobiont. The authors show that stress induces a dysbiotic shift in the microbiota, which is dominated by an outgrowth of Enterobacteriaceae. Building on this, the authors specifically study AIEC colonization under stress, revealing that stress impairs host control of AIEC, by depleting protective IL-22-producing immune cell populations in the intestine.

In general, the manuscript is well written, and the figures are nicely structured and illustrated, making it easy to read and understand. Furthermore, the work provides important and interesting new insights into the role of psychological stress on host-microbiota interactions. Given the fact that disturbances in host-microbiota crosstalk are key driver of IBD, and psychological stress is known to be associated with increased IBD activity, the manuscript addresses important aspects of disease pathology by studying the links between stress and host-microbiota dysbiosis. Therefore the study is likely to be of considerable interest for the readers in the field. In contrast, my major concern is that the strong focus on AIEC makes it difficult to precisely dissect the effects of stress itself, the stress-induced dysbiotic microbiota and the colonization with AIEC on the host, which weakens the overall quality of the paper in its current state.

Response: We thank the reviewer for their positive comments on our manuscript and for their remarks about the impact of our work.

Specific points:

The authors report that stress induces a dramatic microbiota dysbiosis, which is accompanied by defects in barrier functions and increased inflammation. However, the consequences as well as the underlying mechanisms of this direct stress-induced disruption of host-microbiota homeostasis are not comprehensively studied throughout the manuscript. Instead, the authors focus very early on the stress + AIEC colonization model, which introduces another layer of complexity, namely the individual effect of AIEC on the host within an already dysbiotic microenvironment. Therefore, I suggest to separate both models (naïve vs +AIEC) more clearly, which could help to delineate the individual impact of stress and AIEC much better. By this, the authors could first describe the role of stress in creating an (inflammatory) microenvironment that predisposes for pathobiont colonization and outgrowth before showing how AIEC exploits this susceptible state. On the same lines, important questions remain to be answered: What is the individual impact of the stress-induced dysbiotic microbiota on the host? Also, is the stress-induced dysbiotic microbiota needed for AIEC pathogenicity? To answer these questions

experiments with germ-free mice could help to define the specific role of the microbiota for the stress-response.

Response: We thank the reviewer for raising these important points. Regarding the choice of model, although we agree that germ-free mice could be one way to address the points about microbial dysbiosis, we have elected to not introduce this complicated model into our experiments. We are mindful of the developmental abnormalities in the immune system of germ-free mice that we believe would confound the interpretation rather than provide clarity.

Figure 1: What are the dynamics of the microbial dysbiosis and outgrowth of Enterobacteriaceae? Does the dysbiotic microbiota revert back to homeostasis like the AIEC burden shown in Fig. 2b/c?

Response: The reviewer raises an interesting question, and the answer is yes, it does. To address the microbial dynamics, we used 16S sequencing to follow the recovery of the microbiome following stress. Luminal samples were collected from the ileum immediately after stress and 24h after release from stress. We found that 24h after stress, Enterobacteriaceae burdens retract to ~1% abundance, similar to that in control mice. Additionally, many of the bacterial species present in control mice that have reduced abundance following stress rebound following 24h recovery. These findings indicate that, similar to what we observed with AIEC, acute stress causes a transient ileal dysbiosis that can be restored in the post-stress period. We have provided these data below for the reviewer as a measure of good faith, however due to space limitations within the manuscript, we have not included this in the final revision.

Figure 2c: Do repetitive stress and AIEC outgrowth result in some sort of spontaneous intestinal inflammation?

Response: This is an interesting question and one that we intend to address in future follow up work. In our current work, we have chosen to focus on the initial immunological defects associated with the acute stress event rather than defects arising from repeated stress exposure over chronic timescales. The latter is equally interesting to us, but outside the scope of this already sizeable body of work.

Figure 2e: I am wondering why the 16S data from AIEC-colonized naïve and starved animals don't show any abundance of Enterobacteriaceae (except for one mouse per group)? Should the presence of AIEC not also be reflected in the sequencing data?

Response: The AIEC load in the naïve or starved ileum stabilizes around 10^4 cfu/g tissue, as reported in our data. This places AIEC levels below the 1% abundance threshold of taxa set for this analysis. To clarify this point, we have added the following sentence to the results to indicate that Enterobacteriaceae sequencing reads were present in all AIEC colonized control mice but did not rise above the 1% abundance threshold. "In non-stressed control mice, *Enterobacteriaceae* was detected in all mice, but accounted for <1% sequence abundance".

Figure 3c: What is the impact of stress alone (naïve, w/o AIEC colonization) on the barrier genes?

Response: While it was not our focus to fully evaluate stress as an independent variable in a naïve, uncontrived mouse model, as requested by the reviewer we performed this experiment and measured barrier genes by RT-qPCR. As shown below and in new **Supplementary Figure 4**, in uninfected mice exposed to stress alone, expression of genes involved in barrier function were generally decreased compared to unstressed mice, which is consistent with data in the main Figure.

Figure 3e-h: What is the impact of stress alone (naïve, w/o AIEC colonization)?

Response: Again, although our focus was on AIEC as comorbid factor during stress, we did perform the experiment as suggested by the reviewer and have included the naïve mouse data in new Supplementary Figure 4. Similar to barrier genes, uninfected mice exposed to stress alone display some bacterial dissemination to the liver as quantified by plating liver samples on LB agar. However, the frequency of bacteria being present in the liver was lower than in AIEC colonized mice. As such, in line with our original premise, AIEC appears to exacerbate the effect of stress. A similar trend is true for the transcript expression of IL-6, where both naïve and infected stress-exposed mice exhibit increased expression. However, the expression of TLR-4 was not upregulated following stress in naïve mice and remained comparable to controls, likely as a result of the lower barrier disruption shown in the original **Figure 3** and new **Supplementary Figure 4**.

Figure 3i: Is TLR4 inhibition also sufficient to contain Enterobacteriaceae outgrowth (naïve, w/o AIEC colonization)?

Response: This is an interesting question and one we or others may wish to follow up in the future. However, this is outside the scope of the current study where our interest remains focused on the interplay between stress and pathobionts linked to Crohn's disease.

Figure 3j: Again, if this is data from the naïve mice, is this also true for the AIEC colonization model? Moreover, the FACS data don't look very convincing. 80% of all CD45+ ileal lamina propria cells are CD11b+Gr1+ neutrophils?? This appears unrealistic to me. There is also a very high variance within the stress group ranging from 10% to 80%. Maybe the authors could repeat these measurements and include more markers and stain for more cell types, such as T cells, B cells etc. to get a more comprehensive picture.

Response: We agree with the reviewer that the proportion of CD45+ cells that are also CD11b+GR1+ is high, however this result was reproducible across numerous experiments. We repeated the experiment yet again, and saw that stress induces a range of CD45+ cells expressing CD11b+GR1+ in different mice ranging from approximately 10-80% in line with our original data. In addition, we performed immunohistochemistry to confirm the enhanced infiltration of Gr1+ cells (**Fig. 3j**), which demonstrated significantly more Gr1+ cells in stressed mice compared to controls. Thus, we are confident that the data we are presenting in the manuscript is sound.

As requested by the reviewer, we expanded our immunophenotyping of CD90+ cell types after stress. We show that although the overall abundance of CD90+ cells is reduced after stress, the proportion of different T cell subsets was comparable between stress and control groups (**Supplementary Figure 8**), suggesting a generalized depletion of CD90+ cells, rather than a targeted subset. Similarly, we evaluated the frequency of CD45+B220+ cells, which appear to be modestly increased (albeit not significantly) following stress (not shown in manuscript but shown below for reviewer), likely a result of the decreased frequency of CD90+ cells.

Moreover, given the importance of IL-22 in this manuscript, we evaluated the impact of stress on the frequency of the two major IL-22 producing subsets in the intestine, TH17 and ILC3s. Again, we saw a comparable frequency of CD90+ TCRβ+ CD4+ *rorγ*T+ (TH17) cells and CD90+ TCRβ- Lineage- *rorγ*T+ (ILC3s) in control and stress groups. Consistent with our overall conclusions, these data suggest that stress leads to a generalized attrition of CD90+ cells rather than the depletion of a specific cellular subset (**Supplementary Figure 8**).

Figure 4: How is the stress response affecting nutritional immunity in naïve mice (w/o

AIEC colonization)? In this context, how can the outgrowth of endogenous Enterobacteriaceae be explained?

Response: The reviewer raises an important point. To address this concern, we used RT-qPCR and lipocalin-2 ELISA to profile the nutritional immune response during stress of naïve mice. We found that while Lcn-2 is increased in naïve mice following stress, the magnitude of this increase is less than in AIEC colonized mice exposed to stress (**Supplementary Fig. 4**). The expansion of endogenous Enterobacteriaceae is likely explained by the immune cell attrition and possibly other mechanisms to overcome nutritional immunity that were not the focus of the manuscript. To directly address the role of nutritional immunity on AIEC expansion, we used an LPS injection protocol to artificially induce a state of nutritional immunity. Interestingly, we saw that while injecting LPS alone induced robust Lcn2 production similar to stress (**Fig. 5a**), it failed to recapitulate the expansion of AIEC we see following stress (**Fig 5b**). When combined with CD90 depletion, however, LPS injection **was** able to phenocopy the expansion of AIEC seen following stress (**Fig 6e**). These data are consistent with our original conclusions that the combinational effect of nutritional immunity and other immunological changes induced by stress (IL-22 reduction due to CD90+ cell depletion) is required for the beneficial outgrowth of Enterobacteriaceae species following stress.

Figure 5b: What are the total numbers? The authors should also include more inclusion markers for T cells and ILCs. For example: CD90 plus lineage negativity is only a very crude definition of ILCs. Additionally, the authors could use CD127 for ILCs and lineage-defining transcription factors to even define the major T and ILC subsets (Th1, Th2, Th17 and ILC1, ILC2, ILC3). In general: How is stress mediating this depletion of lymphocytes? Is it inducing apoptosis in lymphocytes? This should be experimentally addressed (e.g. by staining for apoptosis markers) and discussed.

Response: Again, the reviewer reiterates an important point that we addressed in the previous comment through a more comprehensive profiling of cells in control and stress exposed mice. As suggested by the reviewer, we also stained for Annexin V to measure the induction of apoptosis following stress. Indeed, at the midpoint (8 hr) of our stress protocol we saw a marked increase of Annexin V⁺ cells (**Fig. 5g**). Importantly, while we observed that the frequency of CD90⁺ cells is similar to control at this time point (Fig. 5f), the frequency of CD90⁺ Annexin V⁺ cells is significantly increased (**Fig 5g**). To address the influence of glucocorticoids on immune cell attrition, we first confirmed the increased presence of corticosterone, which was 2-3-fold higher in stress mice (**Fig 5d**), and that the CD90⁺ population expressed the glucocorticoid receptor, in accordance with published literature (**Fig 5e**). Additionally, pre-treatment with RU486, a glucocorticoid receptor-antagonist, prevented the stress induced reduction in CD45⁺CD90⁺ cells observed at 16 hr (**Fig 5h-i**), but did not impact the production of corticosterone (**Fig 5d**). Together, we believe this adequately demonstrates that glucocorticoid signalling in response to stress is influencing apoptosis of the CD90⁺ population, thereby accounting for the depletion of this cellular population.

Figure 5c/d: Is the cytokine measurement done directly *ex vivo* or after some sort of restimulation? Usually, to detect cytokine expression, restimulation with PMA/Iono or cytokines, e.g. IL-23/IL-1 β for IL-22, is needed. What is exactly depicted in 3c, three individual mice? If yes, why is the background and gating always different? Representative dot plots for IL-17 and IL-22 are missing. The authors should also discriminate between T cells and ILCs here.

Response: The cytokine expression in Figure 5 is done directly *ex vivo* in the absence of stimulation. In our model, we were interested in measuring the difference between *in vivo* activated IL-22 producing cells following starvation or stress, rather than looking at the IL-22 producing potential. Indeed, particularly in stress, we saw a robust IL-22 population by flow cytometry that did not require restimulation. However, as noted above, despite an increase frequency of IL-22 producing cells following stress, the absolute number of cells is significantly reduced. These observations were confirmed using *ex vivo* stimulation with r-IL-23, eliciting the production of IL-22, as determined by ELISA. Indeed, control, but not stress exposed ileal explants significantly increased their production of IL-22 in response to r-IL-23.

Moreover, we have now evaluated the presence of the two major populations capable of IL-22 production in the gut, T_H17 and ILC3s, finding them to be similar in frequency, but not absolute numbers among CD90+ lymphocytes (**Supplementary Figure 8**). Thus, our data remains consistent with a global depletion of CD90+ cells following stress, rather than a specific subset, that is contributing to the loss of IL-22 production.

The original Figure 5c is from three representative mice in the control, stress, and stress + RU486 groups. Unfortunately, our labelling of the graph was not easily understandable. The rows are representative plots for IFN γ , IL-17, and IL-22 for each of the experimental conditions, which is why the gating differs (i.e., gating for IFN γ differs from IL-17 and IL-22). We have clarified this in the other figures in the manuscript but decided to remove old Figure 5c in order to help streamline the paper and highlight the central message of the role of IL-22 in stress-induced dysbiosis.

Figure 5d: The authors show a decrease in IFN γ + and IL-17+ producing CD90+ immune cells upon stress. However, in Figure 3b they show an increased expression of IFN γ , IL-17 etc. in the tissue. Given that T cells and ILCs are the major effector immune cells in the gut, how can these contradictory findings be reconciled?

Response: The reviewer raises a very important point that is central to our main argument and we apologize it was not clearer in the original manuscript. In Figure 3b, cytokine expression was analyzed by RT-qPCR and normalized to the number of cells present by the housekeeping gene *RPLP0*. Because stress results in the depletion of CD90+ cells, RT-qPCR is only revealing how the *limited number of remaining* cells are responding to stress. Similar to the frequency of cytokine positive cells measured by flow cytometry, despite there being an overall significant reduction in the number of CD90+ cells following stress, the remaining surviving cells can respond to the microbial dysbiosis and produce heightened transcript levels of IL-17 and IFN γ

on a per cell basis. We have clarified this in the manuscript to ensure ease of understanding, but overall, the conclusion is that IL-22 drops below a critical threshold for host protection because of the loss of IL-22 producing cells from the population. In the revised manuscript, Figure 5i depicts the absolute number of CD90+ cells, demonstrating this immune attrition is dependent on glucocorticoid signalling.

Figure 5e-h: What is the major source of IL-22 in this model? Are Th17 cells or ILC3s or both mediating protection from AIEC colonization?

Response: This is another important question reiterated by the reviewer. In AIEC colonized mice, IL-22 depletion alone does not affect AIEC colonization levels (**Fig 6c**). It is only under instances of perturbation, such as the induction of barrier breach/nutritional immunity seen following stress, that IL-22 is required to constrain AIEC outgrowth. Importantly, we show IL-22 and CD90+ lymphocytes are required to constrain AIEC outgrowth in stress exposed mice treated with RU486. While it is an interesting question to determine the precise source of this protective IL-22 response, we believe it is outside the scope of this current manuscript and will be an area of follow up.

Figure 6c: It would be very interesting to compare the global ileal transcriptome of stressed AIEC-colonized mice also to the one of stressed naïve mice. Thereby, the effect of stress vs. AIEC colonization could be dissected and AIEC-dependent and AIEC-independent stress-induced pathways could be revealed.

Response: We agree this would be an interesting comparison to help disentangle the effects of stress and AIEC in intestinal inflammation and dysbiosis. Although we believe this is beyond the scope of the current work, we are interested in employing RNA sequencing to explore this question in greater detail in the future as resources allow.

Reviewer #2 (Remarks to the Author):

Psychological stress is known to influence the disease in Crohn's disease (CD) patients. It has been reported that psychological stress can be a driver of gut dysbiosis. However, the effect of psychological stress on CD-associated dysbiosis and its relation to the pathogenesis of disease remain poorly understood. In this manuscript, Shaler et al. investigated the impact of psychological stress on the expansion of CD-associated adherent-invasive *E. coli* (AIEC). The authors demonstrated that restraint stress resulted in the gut dysbiosis accompanied by the expansion of Enterobacteriaceae in the gastrointestinal tract (particularly in the ileum) in SPF mice. Next, the authors used the AIEC colonization model and confirmed that psychological stress promoted persistent colonization by AIEC. The enhanced colonization by AIEC exacerbated DSS-induced colitis in stressed mice. Also, the increased colonization by AIEC impaired the gut barrier integrity and mucosal inflammatory responses.

Psychological stress-induced nutritional immunity, which conferred the competitive outgrowth of AIEC over competing commensals. The authors then focused on IL-22 signaling as a

mechanism by which AIEC expanded in stressed mice. Psychological stress-induced glucocorticoids reduced the number of IL-22-producing lymphocytes (T cells, ILCs), subsequently causing the outgrowth of AIEC. Consistent with this notion, reconstitution of IL-22 signaling restored the gut microbiota and protected mice from colitis.

Linking AIEC expansion and psychological stress is an intriguing concept and potentially crucial for understanding the pathogenesis of CD. The experiments are extensive. However, the merit of the study is significantly dampened by several shortcomings, particularly in the mechanistic depth.

Response: We thank the reviewer for their fulsome critique. We have made important additions to the manuscript to address the mechanistic details that were sought.

Specific comments:

1. AIEC infection model is useful to study the pathogenic role of this particular bacterium. However, the model is somewhat artificial. The authors nicely showed that psychological stress leads to gut dysbiosis with the massive expansion of Enterobacteriaceae (likely dominated by *E. coli*) in Fig. 1. Does the expansion of indigenous Enterobacteriaceae influence the susceptibility to colitis? Can confirm with ASF mice. Were indigenous AIEC-like bacteria (e.g., salmochelin-expressing) are dominated within the Enterobacteriaceae?

Response: This is an important question and relevant to understanding the individual contribution of stress and AIEC to intestinal inflammation. Indeed, our DSS-induced colitis model does indicate that stress in conjunction with AIEC results in significantly greater weight loss compared to AIEC alone, or stress without DSS in the absence of AIEC. Looking at this question in ASF mice could help to clarify how Enterobacteriaceae expansion contributes to DSS-induced weight loss in naïve mice and is an area we are interested in pursuing for follow up studies. For the current study, our interest was focused on comorbid effects of AIEC in conjunction with stress.

2. There was a discrepancy in the regulation of IL-22 signaling. In Fig. 3b, IL-22 and other cytokines (e.g., Th1, Th17-related cytokines) were significantly up-regulated in AIEC/stress mice. However, in Fig. 5, the number of IL-22 (and IL-17A, IFN γ)-producing lymphocytes was reduced in the stressed group. Since the authors claimed that reducing IL-22-producing cells is a crucial mechanism that drives AIEC expansion, this is a critical flaw.

Response: The reviewer raises an important point similar to Reviewer 1 that is central to our main argument. In Figure 3b, cytokine expression was analyzed by RT-qPCR and normalized to the number of cells present by the housekeeping gene *RPLP0*. Because stress results in the depletion of CD90⁺ cells, RT-qPCR is only revealing how the *limited number of remaining cells* are responding to stress. Similar to the frequency of cytokine positive cells measured by flow

cytometry, despite there being an overall reduction in the number of CD90+ cells following stress, the remaining surviving cells are robustly responding to the microbial dysbiosis and producing heightened transcript levels of IL-17 and IFN γ on a per cell basis. We have clarified this in the manuscript to ensure ease of understanding, but overall, the conclusion is that IL-22 drops below a critical threshold for host protection because of the loss of IL-22 producing cells from the population. **Fig 5i** demonstrates this immune attrition and represent the absolute number of CD45⁺CD90⁺ cells. Therefore, due to the global depletion of CD90⁺ cells, the overall cytokine production is lower in stress groups compared to control. We have clarified this in the manuscript to ensure ease of understanding and have removed the frequency of cytokine positive cells from the flow data to provide clarity to this central thesis of the manuscript.

3. The authors claimed that impaired IL-22 signaling by psychological stress is the critical mechanism of gut dysbiosis (AIEC domination). However, upstream and downstream mechanisms of IL-22 signaling defect were insufficiently provided. The authors claimed that glucocorticoid signaling leads to the loss of IL-22-producing cells. Although RU486 treatment restored the number of those cells, the precise mechanism remains unclear. Also, how does IL-22 shapes healthy microbiota was unclear. Since IL-22 is considered as the central pathway that connects psychological stress and gut dysbiosis, more detailed mechanistic validation is required to support the authors' claim.

Response: The reviewer raises points similar to Reviewer 1, which we have addressed experimentally. To investigate the mechanism by which RU486 prevents the loss of the CD90⁺ population, we measured immune cell apoptosis by staining for Annexin V, as suggested by Reviewer 1. We found that 8 hr into stress there was a marked increase in the proportion of CD90⁺AnnexinV⁺ cells compared to controls, indicating that stress leads to immune apoptosis (**Fig 5g**). Importantly, at this time point, the frequency of CD90⁺ cells was similar between control and stress (**Fig 5f**), indicating that this time point is the onset of the cellular attrition period following stress initiation.

These results suggested that glucocorticoids were contributing to the immune cell apoptosis following stress. To further validate the importance of glucocorticoids, we formally demonstrated the enhanced production of corticosterone (**Fig 5d**) and the expression of the glucocorticoid receptor on the surface of the CD90⁺ cell population (**Fig 5e**) and showed that treating mice with the GR-antagonist, RU486, prevented the outgrowth of AIEC typically seen in stress, presumably due to the maintenance of the CD90⁺ population (**Fig 6d**). To confirm this directly, we co-delivered RU486 alongside depleting antibodies for CD90 or IL-22 and showed that blocking the IL-22 response, either by CD90 depletion or by directly neutralizing IL-22, resulted in bacterial expansion to levels similar to those seen during stress regardless of RU486 treatment (**Fig 6d**). These new data provide a mechanistic linkage between glucocorticoid signaling and the loss of CD90⁺ IL-22-producing cells (via apoptosis initiation) that is required for AIEC expansion following stress. To investigate the individual impact of IL-22 even further, we (i) neutralized IL-22 following AIEC colonization either overnight and measured AIEC burden in the ileum (**Fig 6c**), and (ii) neutralized IL-22 over the course of 11 days and measured AIEC in the feces daily (**Supplementary Fig 6**). Consistent with our central thesis, in the absence of

stress, IL-22 neutralization alone was insufficient to cause outgrowth of AIEC. Rather, additional perturbations, including the induction of nutritional immunity, are also required. This central tenet is further supported by our data showing that the induction of nutritional immunity alone (via the delivery of LPS) (**Fig 6e**) or depletion of IL-22 alone (**Fig 6c**) does not lead to AIEC outgrowth to levels seen when both nutritional immunity and loss of IL-22 signaling co-occur (**Fig 6e**). We highlight in the revised manuscript that stress generates this combinatorial effect, which provides the unique environment required for AIEC outgrowth.

4. Although the microbial composition of control and stress mice was nicely analyzed, baseline microbial composition was not shown. It is essential to show the baseline microbiota is standardized in all groups; otherwise, the cage effect cannot be ruled out. At least, a detailed description of how gut microbiota composition was normalized among groups needs to be provided in the method section. Were littermates used? All mice are co-housed or performed mixed-bedding prior to the experiments and then randomized?

Response: We have clarified these details in the methods. Mice arrived in our facility at the same time in the same shipping crate and were treated under the same conditions prior to stress. The mice were caged in the same bedding and our findings were reproducible across numerous distinct batches of mice received at different times. As this study was analyzing the ileal microbiota, it was not feasible to collect a 16S sample prior to stress, as the fecal microbiota is not representative of what is present at the tissue-level.

5. The staining pattern of neutrophils is a little odd. Almost all CD45+ cells in the ileal LP cells are neutrophils in the stress group. The authors should confirm this phenotype by histology (HE staining, immunostaining for neutrophils).

Response: The reviewer raises an important point that was also raised by Reviewer 1. This result was reproducible across numerous experiments. We repeated the experiment yet again, and saw that stress induces a range of CD45+ cells expressing CD11b+GR1+ ranging from approximately 10-80% in line with our original data. In addition, we performed immunohistochemistry to confirm the enhanced infiltration of Gr1+ cells, which demonstrated significantly more Gr1+ cells in stressed mice compared to controls (**Fig. 5i and j**).

6. Based on the experimental procedures provided in Figures, only male mice were used in all animal experiments. Justification needs to be provided on why the authors used only one sex.

Response: Male mice were used in our study because the female estrus cycle leads to the release of progesterone. As the glucocorticoid receptor antagonist RU486 can bind to the progesterone receptor, we wanted to minimize the potential confounding effect that would be present in female animals. This justification has been included in the Methods. All preliminary

results of the study were reproducible in female mice, which showed a similar outgrowth of AIEC following psychological stress.

7. Fig. S6 was related to neither psychological stress nor AIEC colonization. Thus, this data set is not suitable for this study.

Response: We agree with this suggestion and have removed this figure from the study.

Reviewer #3 (Remarks to the Author):

This manuscript reports that acute psychological stress results in a bloom of ileal Enterobacteriaceae and, moreover, makes mice highly prone to colonization by adherent-invasive E. coli, AIEC, which are known to associate with, and thought to promote severity of, Crohn's disease. Accordingly, stress and AIEC synergize to exacerbate DSS-induced colitis. Mechanistically, the phenotype appears to result from altered leukocyte trafficking that impairs IL-22 production by resident lymphocytes.

Wow! This is some of the most striking changes in microbiota that I have ever seen, resulting from a psychological event, albeit a very extreme one. I think the implications of this are quite broad and go well beyond the phenotypic readout used here. The mechanism remains far from understood but, nonetheless, a very solid start has been provided. I think the striking changes in microbiota warrant publication quickly but the mechanistic conclusions could use some better characterization or should at least be toned down. Specific comments are offered.

Response: We thank the reviewer for their positive and supportive comments on our work.

1. Re sampling/analysis of microbiota from various intestine segments, please specify precisely how sample was collected. Is it luminal content that was assayed? Luminal content excluded and only adherent bacteria assayed? Both? I don't really have a concern with what is the specific answer to this query but it needs to be precisely specific so other can reproduce the work.

Response: The reviewer raises an important point, and we thank them for alerting us to the lack of precision. We have remedied this in the Methods. Briefly, luminal samples were collected from the ileum following stress or overnight starvation. The samples for CFU determination were collected as whole tissue which was homogenized and selectively plated for AIEC. For 16s analysis, luminal ileal contents were transferred into a collection tube using forceps. Details have been clarified in the methods.

2. I don't understand how the data is displayed in Fig 1 E-F. In particular, what is the X-axis? Why is bifido on left and fecalbacterium on right. I see empty tick marks. Should there be numbers below them?

Response: Figure 1e,f is a visual representation of the changes in bacterial abundance after stress. The samples are ordered based on the corresponding phylum matching the 16S sequence. As such, x-axis position has no biological meaning and simply allows for spacing of the data points and labels.

3. The ultimate phenotypic readout for impacts of stress and consequently AIEC colonization is exacerbated DSS colitis. This seems slightly incongruous in that stress predominantly impacts the ileum whereas DSS is usually purported to largely affect colon. Hence, it would be helpful to know if stress/AIEC made the SI prone to DSS-induced disease or largely impacted colon disease, or just acted in a general way on a systemically-influenceable parameter like body weight.

Response: In this particular experiment we used DSS to be consistent with previous studies investigating the influence of AIEC or stress on Crohn's disease. While our mechanistic focus for this work was on the ileum, we saw a similar expansion of AIEC in all regions of the intestine. Moreover, we saw similar reduction in expansion in the colon following IL-22-Fc treatment and similar pathways induced (**Supplementary Fig. 7a, b**), suggesting that regions of the lower gut are also susceptible to the impacts of stress. In future work, we are considering employing an ileitis model to have a clearer understanding of the implications of stress in this region specifically.

4. It is clear that overnight withholding of food/water is not sufficient to increase Enterobacteriaceae and proneness to AIEC but is it necessary? Could a mode of stress without starvation suffice? This does not necessarily need to be addressed experimentally at this time but it should be discussed.

Response: We are also curious about other models of psychological stress. In future work we are interested in employing repeated bouts of acute psychological stress to understand how stress progressively disables host defence. We have plans to evaluate this system using an acute variable model of stress, which could include alternation of cold stress, predator stress, caging changes, or other variable stressors. This is a very exciting area of research that has not adequately been explored.

5. The notion that increased TLR4 signaling drives AIEC is interesting as, intuitively, one might have more reasonably presumed that increased TLR4 signaling resulted from AIEC, but certainly both could be true. I suggest this point be discussed. But, in any case, the use of the inhibitor lacks a proper control. TLR4 KO mice are readily-available (on a C57 background or C3HeJ) and this the need for TLR4 in driving stress-induced should be investigated via such mice. These mice can also be used to check the specificity of TAK-242, which is far assured.

Response: The reviewer raised an important point. To address the lack of TAK-242 control we investigated stress-induced outgrowth of AIEC in TLR4 KO mice. Interestingly, following stress, AIEC expanded in TLR4 KO mice to levels similar to those seen in wild type (shown below; not

included in revision). These differences may be due to discrepancies in the microbiome or compensatory mechanisms due to TLR4 deletion, that are beyond the scope of this work to address. Therefore, in the absence of a meticulous understanding of these data, we have elected to remove the TAK-242 data from the revised manuscript.

6. The impairment of IL-22 expression resulting from stress is quite impressive. To what extent is acute loss of IL-22 induction sufficient for AIEC colonization and resulting phenotype? While importing IL-22-KO mice might take awhile, I note their supplier of Fc-IL-22 (Genentech) routinely provides large amounts of neutralizing anti-IL-22 Mab.

Response: This was an excellent point. We obtained neutralizing anti-IL-22 antibody from Genentech and neutralized IL-22 following AIEC colonization and measured AIEC burden in the feces. Consistent with our central thesis, in the absence of stress, IL-22 neutralization alone was insufficient to cause outgrowth of AIEC. Rather, additional perturbations, including the induction of nutritional immunity, are also required. This central tenet is further supported by our data showing that the induction of nutritional immunity alone (via the delivery of LPS) (**Fig 6e**) or depletion of IL-22 alone (**Fig 6c**) does not lead to AIEC outgrowth to levels seen when both nutritional immunity and loss of IL-22 signaling co-occur (**Fig 6e**). We highlight in the revised manuscript that stress generates this combinatorial effect, which provides the unique environment required for AIEC outgrowth.

REVIEWER COMMENTS

Reviewer #1 (Remarks to the Author):

Shaler and Parco et al. have extensively revised their manuscript by adding several experiments that have partially addressed my prior concerns and strengthened their study. The new data are supportive of their conclusions and the new text additions are appropriate.

Unfortunately, the authors missed the chance to better dissect the individual contribution of stress (increase of indigenous Enterobacteriaceae) and AIEC to intestinal inflammation by using GF/ASF mice, as suggested by me and Reviewer 2.

In addition, some minor points remain to be addressed:

- The gating strategy and read-out shown in Suppl. Fig. 8 for Th17 cells and ILC3s needs to be corrected, since ILC3s can also express CD4. Therefore, gating for ILC3s should not exclude CD4+ cells.
- There is a mistake in the labeling of the y-axis in Figure 5c (CD90-PE-Cy7)
- The analysis of IL-22 production or # of IL-22+ CD45+CD90+ lymphocytes (Figure 6A) is still suboptimal, according to the representative dot plot shown in Suppl. Figure 9A (no distinct IL-22+ cell population is visible!).

In my opinion there are two options:

1. The authors repeat these experiments and try to restimulate the cells ex vivo (with e.g. rIL-23, PMA/Iono etc.) to provoke decent IL-22 (IL-17, IFN-g) production. In this case, the authors should highlight the discrepancy between the cytokine response on a single cell level (increased reactivity) and on a global level (reduced reactivity due to the apoptotic loss), which is interesting and would also reconcile the findings in Figure 3b.

OR

2. The authors remove Figure 6A (which is currently not supported by the data) and solely rely on 6B, showing that overall ileal IL-22 production is decreased upon stress and building on the fact that CD45+CD90+ lymphocytes (incl. Th17/ILC3s) are known to be the main producers of IL-22 in the gut.

Reviewer #2 (Remarks to the Author):

The authors adequately addressed my previous concerns.

Reviewer #3 (Remarks to the Author):

The additions, clarifications, and deletions, especially re TLR4 inhibition have improved this manuscript.

I have but one specific minor suggestion. I think a few words need to be added to text to explain that figure 1E shows there is clear treatment-based clustering of microbiota composition not only by PCA but by simple phylogenetic analysis. This point may be obvious to microbiome experts but probably would not be to general Nat Comm readership.

Reviewer #1:

Shaler and Parco et al. have extensively revised their manuscript by adding several experiments that have partially addressed my prior concerns and strengthened their study. The new data are supportive of their conclusions and the new text additions are appropriate.

Unfortunately, the authors missed the chance to better dissect the individual contribution of stress (increase of indigenous Enterobacteriaceae) and AIEC to intestinal inflammation by using GF/ASF mice, as suggested by me and Reviewer 2.

Response: We thank the reviewer for their positive comments. While we agree that the use GF or ASF mice is a logical extension of this work, we maintain it is beyond the scope of this first study and opted to focus on the impact of psychological stress in the context of AIEC infection. We hope to address these suggestions in future work. We have added the following line to the Discussion to highlight the limitations associated with the current study.

“Given that the pathological changes observed following psychological stress are augmented in the presence of AIEC, this work establishes a rationale for future studies to dissect the relative contributions of the microbiome and psychological stress on the gut environment”

In addition, some minor points remain to be addressed:

The gating strategy and read-out shown in Suppl. Fig. 8 for Th17 cells and ILC3s needs to be corrected, since ILC3s can also express CD4. Therefore, gating for ILC3s should not exclude CD4+ cells.

Response: As suggested by the reviewer, we have changed our gating strategy to reflect the inclusion of CD4+ ILCs in the analysis. The revised information is summarized in Suppl. Fig. 8.

There is a mistake in the labeling of the y-axis in Figure 5c (CD90-PE-Cy7)

Response: We appreciate the reviewer noting this oversight. It was the x-axis that was mislabeled, which was corrected to CD45 – PerCP Cy5.5.

The analysis of IL-22 production or # of IL-22+ CD45+CD90+ lymphocytes (Figure 6A) is still suboptimal, according to the representative dot plot shown in Suppl. Figure 9A (no distinct IL-22+ cell population is visible!).

In my opinion there are two options:

1. The authors repeat these experiments and try to restimulate the cells ex vivo (with e.g. rIL-23, PMA/Iono etc.) to provoke decent IL-22 (IL-17, IFN-g) production. In this case, the authors should highlight the discrepancy between the cytokine response on a single cell level (increased reactivity) and on a global level (reduced reactivity due to the apoptotic loss), which is interesting and would also reconcile the findings in Figure 3b.

OR

2. The authors remove Figure 6A (which is currently not supported by the data) and solely rely on 6B, showing that overall ileal IL-22 production is decreased upon stress and building on the fact that CD45+CD90+ lymphocytes (incl. Th17/ILC3s) are known to be the main producers of IL-22 in the gut.

Response: We agree with the reviewer and opted to remove Figure 6a for clarity of data presentation. As suggested, we have included a statement about the likely source of IL-22 being either T_H17 or ILC3s, as they are known to be the main producers of IL-22 in the gut.

“Given the dramatic loss of CD45+CD90+ lymphocytes following stress, these data strongly suggest that stress impairs the IL-22 axis through a reduction in T_H17 and ILC3 cells, as these CD45+CD90+ cell subsets are the main producers of IL-22 in the gut”.

Reviewer #2:

The authors adequately addressed my previous concerns.

Response: We appreciate the reviewer’s time and previous comments.

Reviewer #3:

The additions, clarifications, and deletions, especially re TLR4 inhibition have improved this manuscript.

I have but one specific minor suggestion. I think a few words need to be added to text to explain that figure 1E shows there is clear treatment-based clustering of microbiota composition not only by PCA but by simple phylogenetic analysis. This point may be obvious to microbiome experts but probably would not be to general Nat Comm readership.

Response: We agree with the reviewer that this is a valuable point to highlight. We have added a line to the results section to highlight this observation.

“These data showed a clear treatment-based clustering of microbiota composition not only by PCA but also by phylogenetic cluster analysis (**Fig. 1b-d – top**)”.

REVIEWERS' COMMENTS

Reviewer #1 (Remarks to the Author):

The authors have adressed all my concerns. Thank you!

Reviewer #3 (Remarks to the Author):

I continue to believe the manuscript makes a very solid publication.

REVIEWERS' COMMENTS

Reviewer #1 (Remarks to the Author):

The authors have adressed all my concerns. Thank you!

Response: Delighted to hear it.

Reviewer #3 (Remarks to the Author):

I continue to believe the manuscript makes a very solid publication.

Response: Thank you.